# LoRA3D: Low-Rank Self-Calibration of 3D Geometric Foundation Models

**Ziqi Lu[1,2], Heng Yang[1,3], Danfei Xu[1,4], Boyi Li[1,5], Boris Ivanovic[1], Marco Pavone[1,6], Yue Wang[1,7]**

[1]NVIDIA Research, [2]Massachusetts Institute of Technology, [3]Harvard University, [4]Georgia Institute of Technology, [5]University of California, Berkeley, [6]Stanford University, [7]University of Southern California

`ziqilu@mit.edu, {hengy, danfeix, boyil, bivanovic}@nvidia.com, pavone@stanford.edu, yue.w@usc.edu`

## ABSTRACT

Emerging 3D geometric foundation models, such as DUSt3R (Wang et al., 2024), offer a promising approach for in-the-wild 3D vision tasks. However, due to the high-dimensional nature of the problem space and scarcity of high-quality 3D data, these pre-trained models still struggle to generalize to many challenging circumstances, such as limited view overlap or low lighting. To address this, we propose LoRA3D, an efficient self-calibration pipeline to *specialize* the pre-trained models to target scenes using their own multi-view predictions. Taking sparse RGB images as input, we leverage robust optimization techniques to refine multi-view predictions and align them into a global coordinate frame. In particular, we incorporate prediction confidence into the geometric optimization process, automatically re-weighting the confidence to better reflect point estimation accuracy. We use the calibrated confidence to generate high-quality pseudo labels for the calibrating views and use low-rank adaptation (LoRA) to fine-tune the models on the pseudo-labeled data. Our method does not require any external priors or manual labels. It completes the self-calibration process on a **single standard GPU within just 5 minutes**. Each low-rank adapter requires only **18MB** of storage. We evaluated our method on **more than 160 scenes** from the Replica, TUM and Waymo Open datasets, achieving up to **88% performance improvement** on 3D reconstruction, multi-view pose estimation and novel-view rendering. For more details, please visit our project page.

## 1 INTRODUCTION

Figure 1: Given sparse RGB images, our self-calibration pipeline efficiently specializes a pre-trained 3D foundation model to a target scene to improve its performance for a variety of 3D vision tasks.

Recently, many 3D geometric foundation models have emerged as a potential solution for in-the-wild 3D computer vision tasks such as 3D reconstruction, camera pose estimation and novel view rendering (Wang et al., 2024; Barroso-Laguna et al., 2024; Leroy et al., 2024; Hong et al., 2023). These models, typically enabled by large scale Transformer pre-training, can quickly establish cross-view correspondences and directly regress 3D scene geometry from sparse RGB images. They generalize to a broad range of data and exhibit a strong zero-shot performance on novel tasks.

However, the performance of these pre-trained models can falter under challenging circumstances. For instance, as highlighted in the upper left sub-figure in Fig. 1, DUSt3R's pairwise reconstruction accuracy significantly degrades under low visual overlaps, where certain regions are observed from only a single viewpoint. This decline is rooted in the inherent complexity of 3D geometric inference task, which requires much larger-scale data to fully represent the distribution of real-world 3D data. Unfortunately, the difficulty of annotating in-the-wild 3D data has led to the shortage of high-quality training datasets, limiting the performance of the pre-trained models.

To mitigate the problem, we propose an efficient self-calibration pipeline (Fig. 2), taking only sparse RGB images to specialize pre-trained 3D foundation models to the target scene. Our method requires no manual labeling, camera calibration, or external priors. We only leverage the multi-view consistency of 3D point positions to refine and select pre-trained models' predictions for pseudo labeling. To ensure the pseudo label accuracy, we develop a robust global optimization method to align and refine multi-view predictions while calibrating the prediction confidence. The calibrated confidence strongly correlates with pseudo-label accuracy, allowing us to select high-confidence data for LoRA fine-tuning of the pre-trained model. Our method is tested on 161 test scenes for a variety of 3D vision tasks. It is able to finish the self-calibration process within 5 minutes on a single GPU and deliver performance improvements of up to 88%. The major contributions of our work include (1) the self-calibration pipeline, (2) the robust global optimization method, and (3) the efficient LoRA fine-tuning strategy for DUSt3R self-calibration.

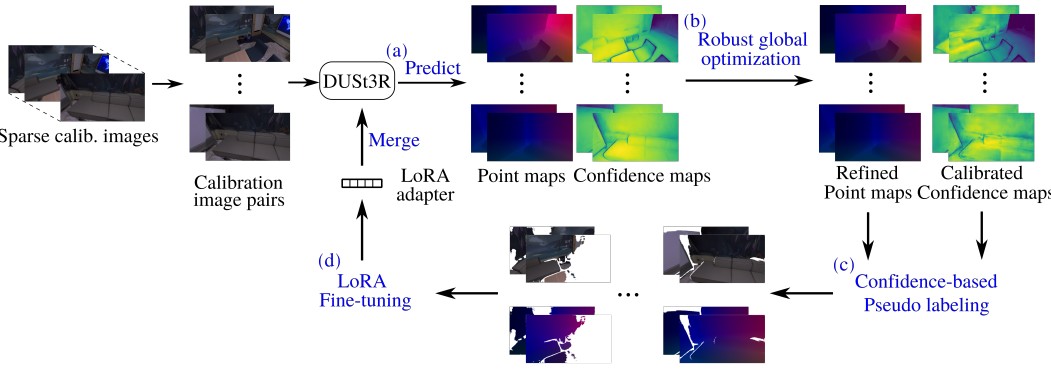

Figure 2: Overview of our self-calibration pipeline. (a) **Predict**: We pair sparse input RGB images and use the pre-trained 3D foundation model to predict per-pair point maps and confidence maps. (b) **Robust Global Optimization**: We apply robust optimization techniques to concurrently refine multi-view point predictions and calibrate prediction confidence. (c) **Confidence-Based Pseudo-Labeling**: Refined point maps with high calibrated confidence are used to generate pseudo-labels on calibration views. (d) **LoRA Fine-Tuning**: Using the pseudo-labeled data, we efficiently fine-tune the pre-trained model with LoRA. While the figure illustrates our method using DUSt3R, our approach generalizes to other 3D foundation models.

## 2  RELATED WORK

### 2.1  FOUNDATION MODEL SPECIALIZATION

Foundation model specialization through fine-tuning or adaptation has become the standard approach to customizing pre-trained foundation models for specific tasks or domains. Various methods have been developed for the specialization of large language models (Brown, 2020; Gururangan et al., 2020), vision-language models (Liu et al., 2024b;a), and vision foundation models (Hu et al.,

2023; Yue et al., 2024). These approaches typically employ parameter-efficient adaptation techniques (Hu et al., 2021; Dettmers et al., 2024; He et al., 2022) to adapt the pre-trained models in either a supervised or unsupervised fashion. However, few works have explored the specialization of 3D geometric foundation models. MASt3R (Leroy et al., 2024) and Spanner3D (Wang & Agapito, 2024) fine-tuned DUSt3R (Wang et al., 2024) to re-purpose it for image matching and incremental reconstruction respectively. Jiang et al. (2024) applied self-training to scale up large reconstruction models (Hong et al., 2023) with real-world images. However, most of these methods still rely on vast amounts of labeled data. In contrast, our method uses only sparse RGB images for self-calibration and requires no ground truth labels.

## 2.2 SELF-SUPERVISED GEOMETRIC PERCEPTION

Self-supervised learning has been successfully applied to a range of geometric perception tasks, including monocular depth prediction (Godard et al., 2019), optical flow prediction (Liu et al., 2019), camera pose estimation (Yang et al., 2021), and structure-from-motion (Zhou et al., 2017), significantly enhancing the performance of pre-trained geometric models. Among these, Yang et al. (2021) is particularly relevant to our approach, as it utilizes robust optimization techniques to generate geometric pseudo-labels for model fine-tuning. However, this method and most others are tailored to adapt smaller-scale pre-trained models for specific tasks. In this work, we extend the pseudo-labeling strategy for self-supervised learning to 3D foundation models. Leveraging the versatility of these models, we can improve their performance on various 3D vision tasks.

## 3 PRELIMINARIES

Our pipeline is primarily tested on DUSt3R (Wang et al., 2024). Below, we provide key details about DUSt3R to give readers the necessary context for understanding our contributions.

### 3.1 DUST3R

As shown in Fig. 2, DUSt3R takes an RGB image pair $(I_i, I_j)$ as input and directly regresses the pixel-wise point maps and confidence maps:

$$(X^{i,i}, C^{i,i}), (X^{j,i}, C^{j,i}) = \text{DUSt3R}(I^i, I^j) \tag{1}$$

Here, $X^{i,i}, X^{j,i} \in \mathbb{R}^{H \times W \times 3}$ are the point maps for view $i$ and view $j$, both expressed in the camera coordinate frame of view $i$, and are regressed up to a unknown scale. Their corresponding confidence maps are denoted as $C^{i,i}, C^{j,i} \in \mathbb{R}^{H \times W}$[1].

### 3.2 RECOVERING CAMERA PARAMETERS

The camera intrinsics can be recovered from the predicted point maps in Eq. 1. Assuming a pinhole camera model with square pixels and principal points at image centers, the camera $i$'s focal length $f_i$ can be estimated by solving the following optimization problem using Weiszfeld algorithm:

$$f_i^* = \arg\min_{f_i} \sum_{p=1}^{HW} C_p^{i,i} \left\| (u_p', v_p') - f_i (X_{p,0}^{i,i}, X_{p,1}^{i,i}) / X_{p,2}^{i,i} \right\| \tag{2}$$

where $(u_p', v_p') = (u_p - W/2, v_p - H/2)$ represents the re-centered image coordinates for pixel $p$.

The relative camera poses are estimated by comparing the predictions for image pair $(I_i, I_j)$ and $(I_j, I_i)$. With point maps $X^{i,i}$ and $X^{i,j}$, we can apply Procrustes alignment (Luo & Hancock, 1999) to estimate the relative pose $T_{i,j} \in \text{SE}(3)$ from camera $i$ to $j$ and the point map scale $\sigma_{i,j}$:

$$(T_{i,j}, \sigma_{i,j})^* = \arg\min_{T_{i,j}, \sigma_{i,j}} \sum_p C_p^{i,i} C_p^{i,j} \left\| \sigma_{i,j} T_{i,j} X_p^{i,i} - X_p^{i,j} \right\|^2 \tag{3}$$

where we omit the homogenization of point maps for brevity.

---

[1] See App. A.1 for details on the training loss of DUSt3R.

## 3.3 Multi-view point map alignment

Given multiple images $\{I_1, I_2, \ldots, I_N\}$ captured in a 3D scene, the multi-view DUSt3R-predicted point maps are aligned to form a global point cloud. Different from bundle adjustment, this alignment is formalized as an 3D-3D-projection-based optimization problem over a connectivity graph $\mathcal{G}(\mathcal{V}, \mathcal{E})$, in which the vertices $\mathcal{V}$ represent the $N$ images and the edges $\mathcal{E}$ represent all image pairs with visual overlaps.

To initialize the optimization parameters, the highest-confidence spanning tree is extracted from the graph. Anchoring the most confident image pair at the origin, the initial estimates of focal lengths, point map scales and relative poses, as derived from Eq. 2,3, are propagated along the tree edges to all $N$ views, yielding initial focal lengths $\{f_i | i = 1, \ldots, N\}$, point maps $\{\chi^i \in \mathbb{R}^{H \times W \times 3}\}$ and image-pair scales $\{\sigma^{(i,j)} \in \mathbb{R}\}$ and poses $\{T^{(i,j)} \in \mathrm{SE}(3)\}$, all expressed in a unified global coordinate frame.

These initial estimates are further refined by minimizing the 3D-3D projection error between the global point maps $\chi$ and the transformed predicted point maps:

$$(\chi, T, \sigma)^* = \arg\min_{\chi, T, \sigma} \sum_{(i,j) \in \mathcal{E}} \sum_{v \in \{i,j\}} \sum_{p=1}^{HW} C_p^{v,i} \left\| \chi_p^v - \sigma^{(i,j)} T^{(i,j)} X_p^{v,i} \right\| \tag{4}$$

Note that the global point maps $\chi_p^v$ can be further re-parameterized via depth back-projection:

$$\chi_p^v = T_v K_v^{-1} D_p (u_p, v_p, 1)^\mathsf{T} = T_v \frac{D_p}{f_v} (u_p', v_p', 1)^\mathsf{T} \tag{5}$$

where $K_v$ and $T_v$ represent the intrinsics and extrinsics for view $v$ and $D_p$ is the depth value for pixel $p$. The optimization problem can therefore be reformulated as:

$$(T, \sigma, f, D)^* = \arg\min_{T, \sigma, f, D} \sum_{(i,j)} \sum_v \sum_p C_p^{v,i} \left\| T_v \frac{D_p}{f_v} (u_p', v_p', 1)^\mathsf{T} - \sigma^{(i,j)} T^{(i,j)} X_p^{v,i} \right\| \tag{6}$$

Here, the per-image-pair poses $T^{(i,j)}$ and per-image poses $T_i$ represent the same transformations but are parameterized separately to allow for additional optimization flexibility. The optimization is solved by a few hundred steps of standard gradient descent. To avoid trivial optimum of $\sigma^{(i,j)} = 0$, $\Pi_{(i,j)} \sigma_{(i,j)} = 1$ is enforced during the optimization.

## 4 Methodology

We aim to adapt a 3D geometric foundation model, such as DUSt3R (Wang et al., 2024), to a target scene using a sparse set of uncalibrated RGB images $\{I_1, I_2, \ldots, I_N\}$. The goal is to enhance the pre-trained model's performance on test images $\{I_{N+1}, I_{N+2}, \ldots, I_{N+M}\}$ from the same scene. Our approach generates compact LoRA adapters, which integrate with the pre-trained model to produce a scene-calibrated model.

### 4.1 Self-calibration pipeline

Fig. 2 shows our self-calibration pipeline. We start by using the pre-trained DUSt3R, as in Eq. 1, to predict point and confidence maps for all calibration image pairs. In challenging conditions, such as under limited camera view overlap, DUSt3R's predictions may include errors and outliers, and the prediction confidence may not precisely reflect the prediction accuracy (See Fig. 3 for an example of overconfident prediction). For this reason, directly relying on predicted confidence for pseudo label selection may hurt the model performance (see Sec. 8 (a,b)).

However, each 3D point in the scene is co-observed by many camera view pairs. We could leverage accurate DUSt3R predictions from well-conditioned, e.g. high-visual-overlap, view pairs to refine and identify inaccurate point map predictions. We therefore develop a robust multi-view point map alignment method (Sec. 4.2) to (1) optimize the point map and (2) calibrate the prediction confidence. We then use the refined point maps and calibrated confidence to pseudo-label the calibration images $\{I_i\}_{i=1}^N$ (Sec. 4.3), after which we fine-tune the pre-trained DUSt3R model using LoRA on the pseudo-labeled data (Sec. 4.4).

## 4.2 ROBUST MULTI-VIEW POINT MAP ALIGNMENT WITH CONFIDENCE CALIBRATION

We develop a robust multi-view point map alignment method by incorporating the prediction confidence into the global optimization in Eq. 6. Specifically, we re-parameterize the confidence term $C_p^{v,i}$ in Eq. 6 as an optimizable weight term $w_p^{v,i}$ to automatically tune each point prediction's contribution to the optimization. While the predicted confidence can be imprecise in challenging cases, it still remains informative for prediction accuracy. Thus we intend to introduce a regularization term that encourages the weights to remain close to the predicted confidence, and also avoid trivial solutions.

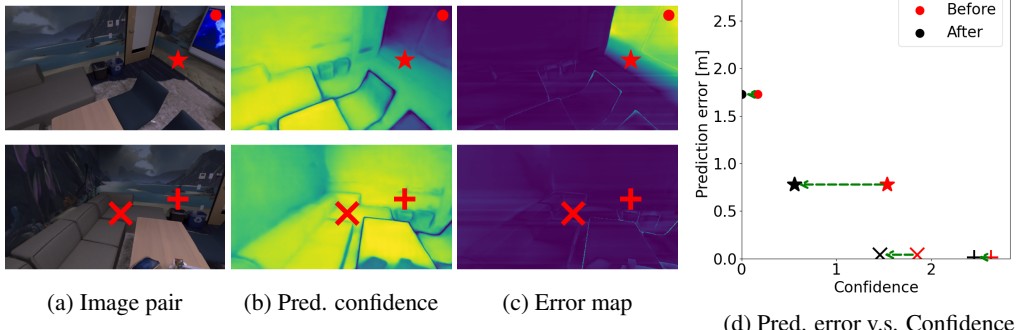

(a) Image pair    (b) Pred. confidence    (c) Error map

(d) Pred. error v.s. Confidence

Figure 3: Pre-trained DUSt3R's (b) prediction confidence and (c) error map on (a) an example image pair: In cases of limited visual overlap, DUSt3R may produce overconfident predictions (★). Our robust multi-view alignment method effectively reduces this overconfidence, maintaining high confidence for accurate predictions (+, ×) and low confidence for outlier predictions (●).

We found our objective to be surprisingly aligned with Geman-McClure robust M-estimator (Geman et al., 1992), which essentially uses a regularization term (an outlier process in robust optimization terminology) to encourage weights to be close to unity in least-squares optimizations. Inspired by this, we designed our regularization term to follow a similar structure. The optimization in Eq. 6 is therefore reformulated as:

$$(T, \sigma, f, D, \mathcal{W})^* = \underset{T,\sigma,f,D,\mathcal{W}}{\arg\min} \sum_{(i,j)} \sum_v \sum_p w_p^{v,i} \|e_p^{v,i}\| + \mu(\sqrt{w_p^{v,i}} - \sqrt{C_p^{v,i}})^2 \qquad (7)$$

where $e_p^{v,i} = T_v D_p(u_p', v_p', 1)^\mathsf{T}/f_v - \sigma^{(i,j)} T^{(i,j)} X_p^{v,i}$ represents the pixel-wise residual error and $\mu$ is a constant hyper-parameter to control the regularization strength.

Rather than updating the weights in the joint optimization loss Eq. 7 via gradient back-propagation, we draw inspiration from the iterative re-weighted least squares approach (Rao & Kreutz-Delgado, 1999) for robust M-estimation, to derive a closed-form weight update rule for fast confidence re-weighting:

$$w_p^{v,i} = C_p^{v,i}/(1 + \|e_p^{v,i}\|/\mu)^2 \qquad (8)$$

(a) Calibrated confidence    (b) Point estimation error    (c) Pixels to pseudo-label    (d) Pseudo labels

Figure 4: Pseudo-labeling with (a) calibrated confidence, which is a good measure of the (b) point estimation accuracy. We select high-calibrated-confidence point predictions as pseudo labels (d) for DUSt3R finetuning.

With this update rule, we can still solve the original optimization problem (Eq. 6) while periodically applying the weight updates. As demonstrated in Appendix A.2, this is equivalent to solving the joint optimization.

The weight update rule can be understood as follows: point predictions with lower residual errors, meaning those that are more consistent with predictions from other image pairs, will maintain confidence similar to the predicted value. In contrast, point predictions that are inconsistent across views will have their confidence significantly reduced. This method effectively minimizes confidence for overly confident predictions, as illustrated in Fig. 3, ensuring that confidence becomes more closely correlated with point estimation accuracy and provides better guidance for global optimization and pseudo-labeling.

### 4.3 PSEUDO LABELING WITH CALIBRATED CONFIDENCE

We use the calibrated confidence and optimized point maps for confidence-based pseudo-labeling. To compute pseudo labels for the calibration image pairs, we need to transform the global optimization results from Eq. 7 to local image-pair coordinate frame. Following Eq. 5, we back-project the optimized depth maps $D_p$ to 3D and transform the points to the image-pair coordinate frame. We then threshold the point estimations with a confidence cutoff $w_{\text{cutoff}}$ and retain the high-confidence ones as pseudo labels. The pseudo labeling rule can be summarized as:

$$\tilde{X}_p^{j,i} = T_i^{*-1} T_j^* \frac{D_p^*}{f_j^*} (u_p', v_p', 1)^\mathsf{T}, \quad \text{where } p \in \{p | w_p^{*j,i} > w_{\text{cutoff}}\} \tag{9}$$

We experimentally found that setting $w_{\text{cutoff}} = 1.5$ works effectively for most test scenes.

Note that our method is naturally robust to dynamic elements in the scene (See Tab. 3). This is because the dynamic points break the multi-view consistency assumption and will be filtered out by pseudo labeling with calibrated confidence.

### 4.4 FINE-TUNING WITH LoRA

On the pseudo-labeled data, we fine-tune the pre-trained DUSt3R with LoRA (Hu et al., 2021) and the same pre-training loss (as Eq. 12). LoRA freezes the pretrained model weights and injects trainable rank decomposition matrices into layers of Transformer architecture, greatly reducing the number of trainable parameters. This (1) improves the runtime- and memory-efficiency of self-calibration and (2) reduces the catastrophic forgetting of the pre-training data.

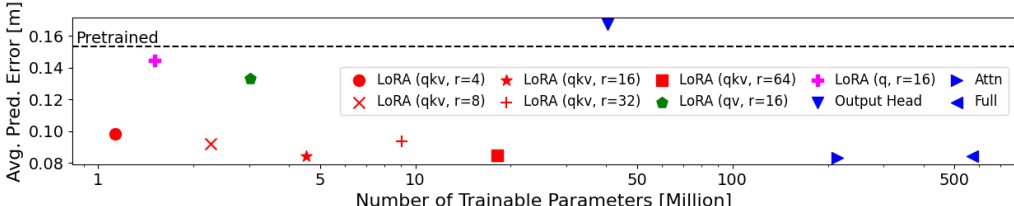

Figure 5: What is the best DUSt3R fine-tuning strategy? We plot the mean prediction errors on test images against the number of trainable parameters for various fine-tuning options on an example test scene (Replica "office0"). We found adapting all attention weights with rank-16 LoRA (i.e. ★) achieves the best trade-off between performance and efficiency on most test scenes.

To find the optimal DUSt3R fine-tuning strategy, we conducted extensive experiments to compare different fine-tuning options across multiple test scenes. Please see Fig. 5 for an example, where we plot the test errors (defined in Sec. 5) against trainable parameter counts for different LoRA and direct fine-tuning strategies. We found adapting all attention weights with rank-16 LoRA often leads to the best trade-off between performance and efficiency. It reduces the number of trainable parameters by more than 99% and has a on-par performance with directly fine-tuning the attention or all weights.

Using rank-16 LoRA, fine-tuning on 10 calibration images converges in under 3.5 minutes with a batch size of 2. Peak GPU memory usage during fine-tuning stays under 20GB, enabling the process to run on a single standard GPU. And each LoRA adapter require less than 18MB of disk storage.

## 5 EXPERIMENTS

We evaluated our method on 161 test scenes for the tasks of 3D reconstruction, multi-view camera parameter estimation and novel view rendering.

**Datasets** We tested our method on *all available test scenes* from the Replica (Straub et al., 2019) and Waymo Open Dataset (Sun et al., 2020), as well as on three test scenes from the TUM RGBD dataset (Schubert et al., 2018) that are most frequently tested in literature. This amounts to a total of 161 test scenes, all of which are distinct from the DUSt3R pre-training scenes.

The Replica dataset comprises eight indoor scenes, each containing 2000 RGB-D images rendered by Sucar et al. (2021). For each scene, the first 1000 RGB images serve as the calibration split and the remaining as the test split. The depth images are not utilized during self-calibration; they are used solely to compute the ground truth point maps. We randomly sample[2] 10 images from the calibration split as the calibration images.

The Waymo Open Dataset has in total 150 test data segments. In each segment, only forward-looking camera images are adopted, where the first 100 form the calibration split and the remaining $\sim$100 images belong to the test split. We sample 10 images from the calibration split for self-calibration.

Please refer to App. A.4 for details about the data splits and tasks for the TUM RGBD dataset.

**Tasks** On the Replica dataset, we evaluate our method for *pairwise* and *multi-view reconstruction* tasks. For pairwise reconstruction, we sample 100 image pairs with visual overlaps from the test split as test images. For multi-view reconstruction, we sample 10 views from the test split.

On the Waymo dataset, we evaluate our method for the tasks of *multi-view camera parameter estimation* and *novel view rendering*. For novel view rendering, we use InstantSplat (Fan et al., 2024), which adopts DUSt3R-predicted point cloud and camera parameters as initialization, to train 3D Gaussian Splatting (3DGS) models and render novel-view images. From the test split, we select every 10th images (i.e. 0th, 10th, 20th, $\cdots$) for camera pose estimation evaluation and InstantSplat training. Images at indices 5, 15, 25, $\cdots$ are used as novel views to evaluate the InstantSplat renders. Note that although our method is robust to dynamic environments, InstantSplat relies on the static world assumption to train 3DGS. We therefore selected segments 10084, 10649, and 10802 – that are mostly static – from the first 10 test segments for the novel view rendering evaluation.

**Baselines** The two most important comparison models for our self-calibrated DUSt3R (**DUSt3R-Self-Calib**) are the pre-trained DUSt3R (**DUSt3R-Pretrain**) and the fine-tuned DUSt3R on ground truth point maps of calibration image pairs (**DUSt3R-GT-FT**). The ground-truth fine-tuned model is considered as the upper limit of ours, serving as an oracle model.

Both methods are evaluated in most tests, with the exception of the TUM dataset, where no ground truth depth is available. In this case, we use noisy depth measurements for fine-tuning, referred to as **DUSt3R-Depth-FT**. On the Waymo dataset, we use the high-quality Lidar point clouds for ground-truth-based fine-tuning. For a fair comparison, the training hyperparameters for ground-truth and depth-based fine-tuning are kept consistent with those used in our method.

For *multi-view stereo reconstruction*, we also use **COLMAP** (MVS)(Schonberger & Frahm, 2016; Schönberger et al., 2016), **FlowMap** (Smith et al., 2024) and **MASt3R** (Duisterhof et al., 2024) as baselines, all of which perform dense reconstructions with un-calibrated images.

*COLMAP* (MVS) is a standard SfM and MVS pipeline for which we adopt the default setups.

*FlowMap* is a differentiable SfM model for RGB videos. It relies on optical flow and point tracking algorithms to bootstrap its scene parameter optimization process.

*MASt3R* re-purposes DUSt3R for image matching. Beyond 3D point regression, it establishes accurate cross-view correspondences, and leverages both 3D-3D and 2D-3D correspondences for global point map alignment. We adopt the pre-trained MASt3R with default hyper-parameters.

For fair comparison in multi-view reconstruction, we use the same global optimization method for DUSt3R and its variants (as detailed in Sec. 3.3), and retain all points for evaluation without applying confidence-based filtering to DUSt3R and MASt3R reconstructions.

---

[2] Unless specified otherwise, all random sampling use seed=0 for re-producibility.

For *multi-view camera parameter estimation*, we use **COLMAP**, **RelPose++** (Lin et al., 2024), **PoseDiffusion** (Wang et al., 2023), **RayDiffusion** (Zhang et al., 2024), **FlowMap** (Smith et al., 2024) and **MASt3R** (Duisterhof et al., 2024) as the additional baselines.

*RelPose++* uses a pairwise scoring network and a multi-view reasoning transformer to predict multi-view camera poses.

*PoseDiffusion* develops a diffusion-based bundle adjustment method to estimate multi-view camera parameters. A geometry-guided sampling (GGS) scheme is applied to enforce epipolar constraints across views. We adopt the GGS-enabled PoseDiffusion with the CO3Dv2 checkpoint.

*RayDiffusion* re-parameterize cameras as rays and applies a ray diffuser network to denoise camera rays and recover camera parameters.

All three methods above are pre-trained on domain-specific data without further fine-tuning.

For *novel view rendering*, we use the pre-trained, self-calibrated and finetuned DUSt3R for InstantSplat's 3DGS initialization, and the different variants for InstantSplat are refered to as: **InstantSplat** (Fan et al., 2024), **InstantSplat-Self-Calib**, **InstantSplat-GT-FT**.

**Evaluation Metrics** For DUSt3R *pairwise reconstruction*, we use the *average point prediction error* as the evaluation metric. This is the average Euclidean distance calculated between the predicted and ground truth point maps within local image-pair coordinate frames, with predicted maps normalized and re-scaled to align with ground truth.

We assess *multi-view stereo reconstructions* based on *accuracy* and *completeness* relative to the ground truth Replica mesh models (Straub et al., 2019). Accuracy measures the average distance of reconstructed points to their nearest mesh points, while completeness measures the average distance of mesh points to their nearest reconstructed points. Following Zhu et al. (2022), we exclude mesh parts invisible to the test images.

The *multi-view camera parameter estimations* are evaluated with the *absolute trajectory error* (ATE) and the *average focal length estimation error* (AFE).

The quality of novel view renders is assessed with *Peak Signal-to-Noise Ratio* (PSNR), *Structural Similarity Index Measure* (SSIM), and *Learned Perceptual Image Patch Similarity* (LPIPS).

## 5.1 RESULTS

For the Replica dataset, we report pairwise and multi-view reconstruction results in Tab. 1 and Tab. 2. Compared to the pre-trained DUSt3R, our method reduces point prediction errors by up to 38% and reconstructs models that are up to 61% more accurate and 41% more complete. As Fig. 6 shows, our approach is particularly effective at reducing outlier point predictions, thanks to the multi-view consistent pseudo labels [3]. The remaining performance gap compared to the fine-tuned DUSt3R is attributed to differences in data size and label accuracy.

Since cameras in Replica are mostly facing inward, it is easier for COLMAP and MASt3R to establish accurate cross-view feature matches, resulting in better accuracy and completeness on certain test scenes, even surpassing the fine-tuned DUSt3R. On the other hand, FlowMap struggles due to the discontinuous calibration images, which disrupt the optical flow and point tracking it relies on.

Table 1: **Quantitative evaluation** of **pairwise reconstructions** on the Replica dataset. We report the average point prediction errors (cm) for direct DUSt3R predictions.

| Methods | office0 | office1 | office2 | office3 | office4 | room0 | room1 | room2 |
|---|---|---|---|---|---|---|---|---|
| DUSt3R-Pretrain | 14.29 | 11.02 | 14.03 | 15.44 | 14.96 | 13.11 | 27.99 | 16.82 |
| **DUSt3R-Self-Calib** | **8.84** | **9.38** | **11.05** | **14.41** | **13.92** | **13.02** | **19.88** | **13.65** |
| DUSt3R-GT-FT | 7.12 | 7.95 | 10.55 | 12.88 | 12.29 | 9.27 | 17.40 | 12.58 |

The multi-view camera parameter estimation results on the Waymo dataset are presented in Tab. 3. Compared to the pre-trained DUSt3R, our method reduces camera trajectory estimation errors by up to 88% and focal length estimation errors by up to 79%. Out of 150 total test scenes (detailed in

---

[3]Please check out A.9 for qualitative comparison of the reconstructions with the ground truth mesh model.

Table 2: **Quantitative evaluation** of **multi-view reconstructions** on the Replica dataset. We report the accuracy (**Acc.** [cm] ↓) and completeness (**Comp.** [cm] ↓) of 3D reconstructions against the ground truth meshes. Please refer to App.Tab. 9 for the remaining results omitted due to space limit.

| Methods | office0 | | office1 | | office2 | | office3 | | office4 | |
|---|---|---|---|---|---|---|---|---|---|---|
| | Acc. | Comp. | Acc. | Comp. | Acc. | Comp. | Acc. | Comp. | Acc. | Comp. |
| DUSt3R-Pretrain | 5.22 | 6.78 | 9.21 | 9.27 | 6.57 | 8.35 | 8.43 | 11.89 | 12.97 | 15.89 |
| **DUSt3R-Self-Calib** | 4.43 | 6.08 | **3.56** | 5.48 | 4.75 | **6.89** | 6.60 | 11.00 | 7.81 | 12.22 |
| DUSt3R-GT-FT | 3.51 | 5.29 | 3.26 | 5.53 | 3.93 | 6.72 | 4.02 | 7.42 | 5.53 | 11.25 |
| COLMAP (dense) | **2.61** | 89.87 | 58.15 | 158.83 | 4.87 | 194.16 | **5.51** | 162.53 | 6.42 | 120.84 |
| FlowMap | 51.78 | 152.05 | 142.81 | 107.17 | 24.16 | 189.86 | 19.58 | 248.16 | 15.34 | 153.64 |
| MASt3R | 4.69 | **6.05** | 3.92 | **4.87** | **4.09** | 7.39 | 7.17 | **9.42** | **4.87** | **11.52** |

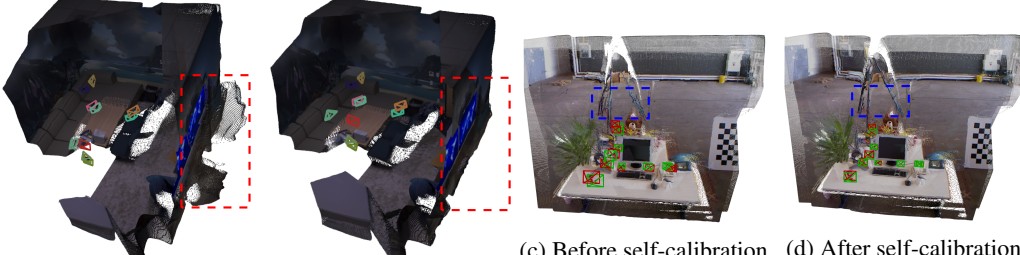

(a) Before self-calibration   (b) After self-calibration   (c) Before self-calibration   (d) After self-calibration

Figure 6: **Qualitative results** on the Replica (a,b) and TUM (c,d) datasets. After DUSt3R self-calibration, we observe much fewer outlier points in the reconstruction of the Replica scene "office0". On the TUM scene "fr2_xyz", the green and red frustums represent the estimated and ground truth cameras respectively. The camera pose estimates are made more accurate by self-calibration.

App.A.11), our approach successfully improves camera parameter estimation results on 116 scenes. Most failures occur when the test vehicle remains mostly static for the test images (e.g. segment-10488). This degenerate case expect the model to predict zero relative pose across views and fails to distinguish different methods.

By comparison, COLMAP fails, and MASt3R shows degraded performance on Waymo due to the presence of dynamic objects and the larger baselines between forward-facing cameras, which make feature matching more difficult. FlowMap still struggles due to the abrupt visual changes across views. RelPose++, PoseDiffusion and RayDiffusion, without domain- or scene-specific training, fails to provide accurate estimates on the out-of-distribution data.

The novel view rendering results on Waymo are presented in Tab. 4 and Fig. 7. Our method effectively reduces floating artifacts in the optimized 3DGS, resulting in quality improvements of up to 0.97 dB in PSNR, 0.09 in SSIM, and 0.04 in LPIPS.

The quantitative results on the TUM dataset are presented and analyzed in App. A.4.

Table 3: **Quantitative evaluation** of **camera parameter estimates** on **Waymo** Open Dataset. We report the absolute trajectory error (**ATE** (m) ↓) and average focal error (**AFE** (%) ↓) for test camera views. Please see App. A.11 for full evaluation results on 150 test scenes.

| Methods | segment-10084 | | segment-10149 | | segment-10649 | | segment-10802 | | segment-10980 | |
|---|---|---|---|---|---|---|---|---|---|---|
| | ATE | AFE | ATE | AFE | ATE | AFE | ATE | AFE | ATE | AFE |
| DUSt3R-Pretrain | 0.79 | 2.19 | 0.84 | 3.08 | 0.95 | 2.84 | 0.35 | 1.60 | 0.80 | 1.19 |
| **DUSt3R-Self-Calib** | 0.37 | **0.61** | **0.25** | 2.14 | **0.49** | 2.54 | **0.35** | 1.08 | **0.09** | 0.69 |
| DUSt3R-GT-FT | 0.20 | 0.17 | 0.17 | 1.54 | 0.29 | 1.73 | 0.39 | 0.55 | 0.13 | 0.49 |
| COLMAP | Fail | Fail | Fail | Fail | Fail | Fail | Fail | Fail | Fail | Fail |
| Flowmap | **0.31** | 3.97 | 66.62 | **1.80** | 36.44 | 13.74 | 21.16 | **0.44** | 65.17 | **0.66** |
| PoseDiffusion | 19.43 | 25.07 | 16.76 | 49.18 | 20.19 | **2.26** | 13.61 | 23.74 | 18.19 | 31.04 |
| RayDiffusion | 17.34 | 85.65 | 16.91 | 80.69 | 18.59 | 85.09 | 12.77 | 81.44 | 19.12 | 85.00 |
| RelPose++ | 14.80 | - | 16.20 | - | 13.69 | - | 12.92 | - | 13.55 | - |
| MASt3R | 2.85 | 11.87 | 1.35 | 24.92 | 0.65 | 20.53 | 1.26 | 24.75 | 1.61 | 6.59 |

Table 4: **Quantitative evaluation** of **novel view renders** on the **Waymo** open dataset

| Methods | Segment-10084 | | | Segment-10649 | | | Segment-10802 | | |
|---|---|---|---|---|---|---|---|---|---|
| | PSNR ↑ | SSIM ↑ | LPIPS ↓ | PSNR ↑ | SSIM ↑ | LPIPS ↓ | PSNR ↑ | SSIM ↑ | LPIPS ↓ |
| InstantSplat | 21.45 | 0.67 | 0.33 | 22.45 | 0.72 | 0.30 | 25.94 | 0.79 | 0.24 |
| **InstantSplat-Self-Calib** | **22.42** | **0.76** | **0.29** | **22.81** | **0.77** | **0.27** | **26.36** | **0.81** | **0.22** |
| InstantSplat-GT-FT | 22.64 | 0.75 | 0.27 | 23.07 | 0.78 | 0.27 | 26.43 | 0.81 | 0.22 |

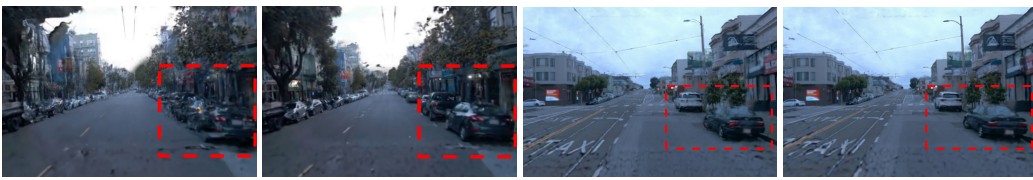

(a) Before self-calibration (b) After self-calibration (c) Before self-calibration (d) After self-calibration

Figure 7: Novel view renders by InstantSplat (Fan et al., 2024) before and after DUSt3R self-calibration on Waymo Open Dataset.

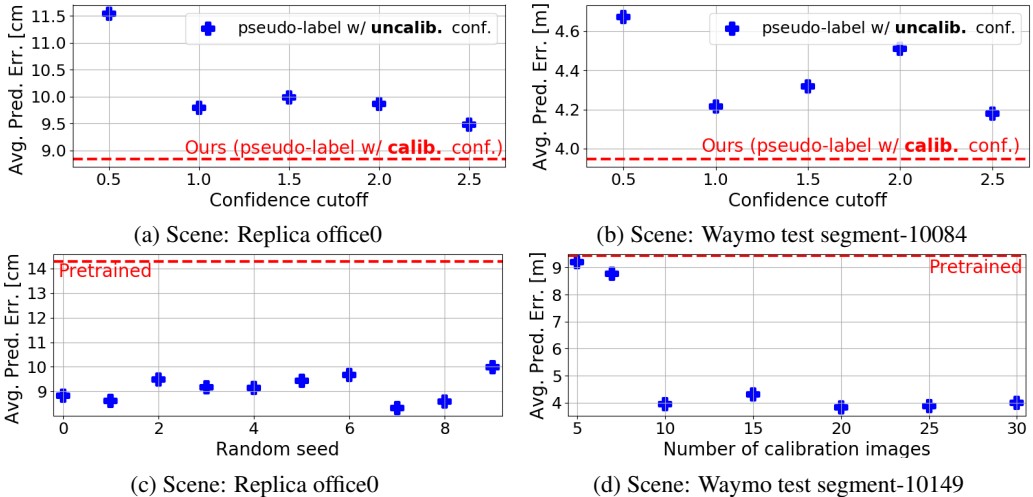

(a) Scene: Replica office0      (b) Scene: Waymo test segment-10084

(c) Scene: Replica office0      (d) Scene: Waymo test segment-10149

Figure 8: Ablation study: (a, b) Pseudo-labeling with *un-calibrated confidence* hurt the model performance. (c) Our method maintains consistent performance across varying random seeds used for calibration image sampling. (d) Our method's performance improves with more calibration images and saturates after around 10.

## 5.2 MASt3R SELF-CALIBRATION

Our pipeline is not limited to the specialization of DUSt3R. We show in App. A.5 that the same idea applies to MASt3R.

## 5.3 ABLATION STUDY

**Un-calibrated confidence for pseudo labeling** Directly using the prediction confidence for pseudo-labeling could harm the model performance. As Fig. 8 (a, b) shows, thresholding refined point predictions with un-calibrated confidence leads to consistent under-performance of the self-calibrated model, regardless of the confidence cutoff value.

**Varying random seed** Our method works with casually captured RGB images and does not rely on carefully selected calibration images to succeed. As shown in Fig. 8 (c), our method maintains consistent performance across varying random seeds used for calibration image sampling.

**The number of calibration images** As shown in Fig. 8 (d), using few calibration images (e.g., fewer than 10) limits our method's performance due to an insufficient number of view pairs to enforce multi-view consistency in global optimization and a limited training data size. We also observe that the performance typically saturates after around 10 calibration images.

**The size of calibration split & Multi-scene concurrent self-calibration** See A.6 & A.7 for details.

## 6 CONCLUSION

Our self-calibration pipeline specializes 3D geometric foundation models to target scenes in a highly time- and memory-efficient manner. It boosts pre-trained model performance by up to 88% across diverse datasets and 3D vision tasks. However, in certain cases, the self-calibrated model still falls short of competing methods due to the inherent difficulty of 3D geometric inference.

## 7 ETHIC STATEMENT

Our work utilizes publicly available datasets that adhere to strict ethical guidelines. These datasets ensure that personally identifiable information, such as human faces and vehicle license plates, is either blurred or anonymized to safeguard privacy. Our work does not engage with human subjects or introduce concerns regarding fairness or potential misuse. We are fully committed to maintaining ethical integrity throughout the development and application of our methods.

## 8 REPRODUCIBILITY STATEMENT

To ensure the reproducibility of our results, we have taken the following steps: (1) A detailed explanation of our method in Sec. 4, along with implementation details provided in Appendix A.3. (2) All datasets used in our experiments are publicly accessible. (3) Comprehensive experimental results for 161 test cases are included in Sec. 5 and further detailed in Appendix A.4, A.10, and A.11.

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

## A  APPENDIX

### A.1  DUSt3R TRAINING LOSS

DUSt3R adopts a 3D-regression-based training objective, which computes the pixel-wise Euclidean distance between the predicted and ground truth point maps $\bar{X}^{i,i}, \bar{X}^{j,i}$ over pixels $\mathcal{P}^{i,i}, \mathcal{P}^{j,i} \subseteq \{1 \ldots H\} \times \{1 \ldots W\}$ where ground truth is available. For example, the regression loss for a valid pixel $p \in \mathcal{P}^v$ on image $v$ from image pair $(i,j)$ is computed as :

$$\ell_{\text{regr}}(v,p) = \|\frac{1}{z}X_p^{v,i} - \frac{1}{\bar{z}}\bar{X}_p^{v,i}\|, \quad \text{where } v \in \{i,j\} \tag{10}$$

Here, $z$ and $\bar{z}$ are normalization factors used to resolve the scale ambiguity between predicted and ground truth point maps. They are defined as the average distance of all valid points in image pair $(I_i, I_j)$ to the origin:

$$z(X^{i,i}, X^{j,i}) = \frac{1}{|\mathcal{P}^i| + |\mathcal{P}^j|} \sum_{p \in \mathcal{P}^i} \|X_p^{i,i}\| + \sum_{p \in \mathcal{P}^j} \|X_p^{j,i}\| \tag{11}$$

The final training loss for DUSt3R is the confidence-aware loss from Wan et al. (2018), defined as:

$$\mathcal{L}_{\text{conf}} = \sum_{(i,j) \in \mathcal{E}} \sum_{v \in \{i,j\}} \sum_{p \in \mathcal{P}^v} C_p^{v,i} \ell_{\text{regr}}(v,p) - \alpha \log C_p^{v,i} \tag{12}$$

This loss enables the model to learn confidence predictions that are correlated with the regression accuracy. The second term acts as a regularization component, where $\alpha$ is a hyperparameter that controls the strength of the regularization.

### A.2  DERIVATION OF WEIGHT UPDATE RULE

Starting from the joint optimization Eq. 7, we first minimize over the weight terms $w_p^{v,i}$ to eliminate them from the joint optimization:

$$(T, \sigma, f, D)^* = \arg\min_{T,\sigma,f,D,} \sum_{(i,j)} \sum_{v} \sum_{p} \min_{w_p^{v,i}} \left[ w_p^{v,i} \|e_p^{v,i}\| + \mu(\sqrt{w_p^{v,i}} - \sqrt{C_p^{v,i}})^2 \right] \tag{13}$$

In order to find the global minimizers $w_p^{v,i}{}^*$, we take the gradient $g_w$ of the above objective function with respect to $w_p^{v,i}$:

$$g_w = \|e_p^{v,i}\| + \mu \left[ 1 - \sqrt{C_p^{v,i}/w_p^{v,i}} \right] \tag{14}$$

We observe that the gradient $g_w$ is continuous and monotonic for $w_p^{v,i} > 0$. Also, if $w_p^{v,i} \to 0$, then $g_w \to -\infty$, and if $w_p^{v,i} \to +\infty$ then $g_w = \|e_p^{v,i}\| + 1 > 0$. Therefore, there exists an unique global minimizer $w_p^{v,i}{}^*$ at which the gradient $g_w$ evaluates to zero. Setting $g_w = 0$, we can solve for $w_p^{v,i}{}^*$ as :

$$w_p^{v,i}{}^* = C_p^{v,i}/(1 + \|e_p^{v,i}\|/\mu)^2 \tag{15}$$

This gives us the same weight update rule in Eq. 8. During global optimization, we alternate between the gradient descent step to optimize geometric parameters: $T, \sigma, f, D$ and the weight update step to set the weight terms to their global minimizers. This helps us down-weight the confidence for overconfident point predictions, and make the optimization more robust on challenging circumstances.

### A.3 IMPLEMENTATION DETAILS

Our pipeline is implemented with PyTorch and all our experiments are conducted on a NVIDIA 3090 GPU.

For robust global point map alignment, we set the regularization coefficient $\mu$ to 0.01. We minimize the optimization loss by running 300 steps of gradient descent using the Adam optimizer with a learning rate of 0.01, applying the closed-form weight update Eq. 8 every 10th gradient descent step. Additionally, we exclude points with prediction confidence below 0.5 by setting their weights to zero, preventing them from participating in the optimization process.

For confidence-based pseudo labeling, we use a cofidence threshold of 1.5 for all test scenes.

For LoRA fine-tuning, we resize all calibration images to the pre-training resolution of (512, 384). During fine-tuning, we optimize the LoRA weights over 10 epochs (without warmup) using the AdamW optimizer with a batch size of 2. A cosine decay learning rate scheduler is employed, with a base learning rate of 0.001 and a minimum learning rate of 0.00001 for most test cases.

### A.4 EVALUATION RESULTS ON TUM RGBD DATASET

We evaluated our method for *multi-view camera parameter estimation* on TUM test scenes: "fr1_desk", "fr2_xyz" and "fr3_office" in the TUM dataset. In each scene, the first 500 RGB images are reserved as the calibration split, and the remaining 92/2897/2015 images form the test split. At test time, we sample 10 images from the calibration split for self-calibration and 10 images from the test split to assess pose estimates.

Table 5: **Quantitative evaluation** of **camera parameter** estimates on the **TUM** RGBD dataset.

| Methods | fr1_desk | | fr2_xyz | | fr3_office | |
|---|---|---|---|---|---|---|
| | ATE (cm) | AFE (%) | ATE (cm) | AFE (%) | ATE (cm) | AFE (%) |
| DUSt3R-Pretrain | 0.91 | 8.02 | 3.89 | 14.84 | 3.28 | 1.95 |
| **DUSt3R-Self-Calib** | 0.62 | 8.32 | 1.24 | 7.67 | 3.10 | 1.81 |
| DUSt3R-Depth-Calib | 0.68 | 7.63 | 1.23 | 4.71 | 4.12 | 1.78 |
| COLMAP | **0.51** | 3.87 | **0.97** | **4.92** | 1.98 | 4.71 |
| RelPose ++ | 10.05 | - | 34.59 | - | 96.91 | - |
| PoseDiffusion | 11.90 | 27.18 | 34.32 | 37.30 | 92.61 | 1.28 |
| RayDiffusion | 10.95 | 72.30 | 35.63 | 12.12 | 100.56 | 3.52 |
| FlowMap | 8.03 | 39.58 | 14.30 | 84.21 | 32.67 | 24.95 |
| MASt3R | 0.56 | **3.32** | 1.74 | 10.88 | **0.92** | **1.24** |

The results of multi-view camera parameter estimation on TUM are reported in Tab. 5 and Fig. 6 (c, d). Our self-calibrated DUSt3R achieves up to 68% more accurate camera pose estimates and up

to 48% more accurate focal length estimates compared to the pre-trained baseline. The slight drop in focal length estimation accuracy for the "fr1_desk" sequence is likely due to overfitting on the calibration images.

Using depth measurements as the supervision signal for DUSt3R fine-tuning appears to be unreliable. DUSt3R-Depth-Calib under-performs our method in "fr1_desk" and even falls short of the pre-trained model in "fr3_office." This is the result of the noise and outliers present in the depth images.

Similar to Replica, COLMAP and MASt3R tends to perform well on the inward-facing cameras in TUM. RelPose++, PoseDiffusion, RayDiffusion and FlowMap fail to provide reasonable estimates of camera parameters.

## A.5 MASt3R SELF-CALIBRATION

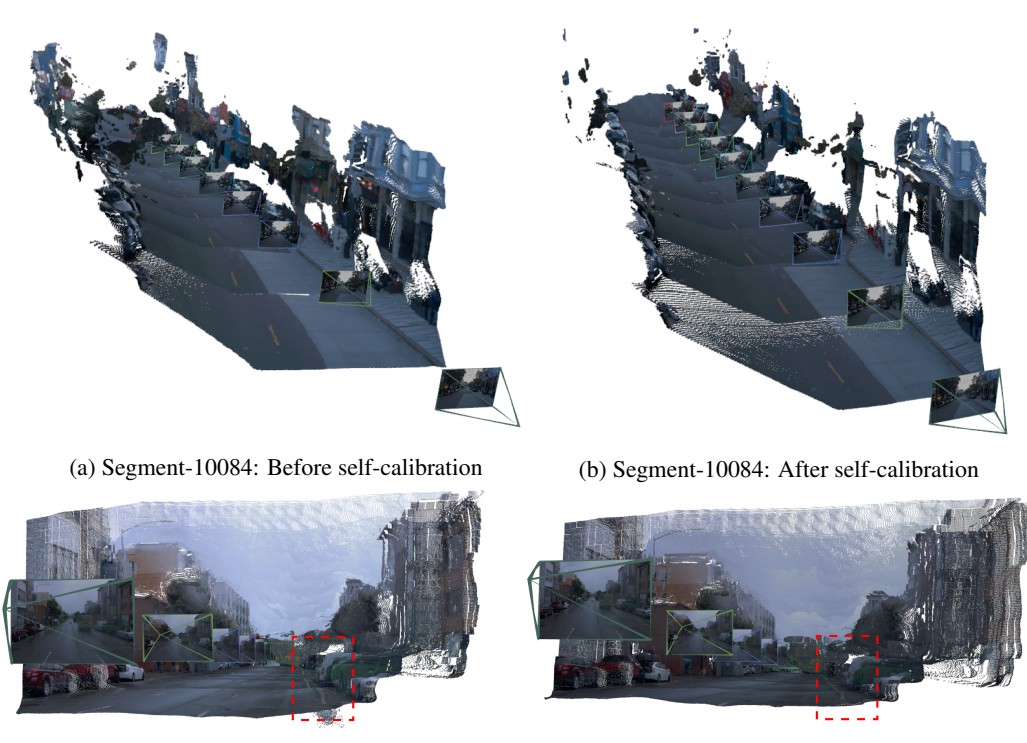

(a) Segment-10084: Before self-calibration

(b) Segment-10084: After self-calibration

(c) Segment-10149: Before self-calibration

(d) Segment-10149: After self-calibration

Figure 9: **Qualitative results of MASt3R self-calibration** on the Waymo test segments 10084 and 10149. After self-calibration, MASt3R produces more accurate camera pose estimates. For segment-10084, the road's center lines exhibit better alignment along a straight path. For segment-10149, the roadside parking lines align more accurately along a straight line.

We show that our method generalizes to MASt3R (Leroy et al., 2024; Duisterhof et al., 2024) self-calibration.

MASt3R builds upon DUSt3R and re-purposes it for more precise image matching. In addition to pairwise point and confidence predictions, MASt3R also generates dense local feature maps to establish 2D feature correspondences across images. Similar to DUSt3R, MASt3R employs a global optimization procedure to align and optimize the local per-pair point predictions within a global coordinate frame. Leveraging both 3D-3D and 2D-3D correspondences, MASt3R first minimizes a 3D projection loss as in Eq. 6 to optimize the image pair scales and camera poses, followed by minimizing a 2D re-projection loss to optimize the camera focal lengths and depth maps.

For MASt3R self-calibration, we also utilize the global optimization results to generate pseudo labels. For simplicity, we directly apply all point predictions across all calibration views for pseudo

labeling, without employing confidence-based filtering. We maintain the use of rank-16 LoRA and the same pre-training loss for fine-tuning MASt3R.

Table 6: **Quantitative evaluation for MASt3R self-calibration** on the Replica "office0", TUM "fr2_xyz" and Waymo test segments 10084 and 10149.

| Scene | Replica office0 | | TUM fr2_xyz | | Waymo seg.-10084 | | Waymo seg.-10149 | |
|---|---|---|---|---|---|---|---|---|
| Metric | Acc.[cm] | Comp.[cm] | ATE[cm] | AFE[%] | ATE[m] | AFE[%] | ATE[m] | AFE[%] |
| MASt3R-Pretrain | 4.69 | 6.05 | 1.74 | 10.88 | 2.85 | **11.87** | 1.35 | 24.91 |
| **MASt3R-Self-Calib** | **4.61** | **6.02** | **1.60** | **10.48** | **1.21** | 16.71 | **1.19** | **22.97** |
| MASt3R-GT-FT | 4.25 | 5.83 | - | - | 1.05 | 9.87 | 1.21 | 24.30 |

We evaluated our pipeline on four test scenes: Replica "office0", TUM "fr2_xyz", and Waymo test segment-10084 and segment-10149. The self-calibrated MASt3R was assessed on multi-view reconstruction and multi-view camera parameter estimation using the same evaluation metrics against the pre-trained baseline and the ground-truth fine-tuned model. The results are reported in Tab. 6 and Fig. 9.

We observe improved performance of MASt3R across various tasks and datasets compared to the pre-trained model, though the gains are mostly marginal, likely due to the presence of unfiltered noise and outliers in the pseudo-labeled data. In future work, we plan to calibrate MASt3R by incorporating calibrated-confidence-based pseudo labeling.

## A.6 ABLATION STUDY ON THE SIZE OF CALIBRATION SPLIT

In addition to the number of calibration images, the size of the calibration split plays an important role in self-calibration because it affects the diversity of viewpoints among the calibration images. As shown in Fig. 10, when we limit the calibration split to only the first 10 to 50 images in a test scene, we observe a decrease in the performance of the self-calibrated model.

This decline occurs primarily because the sampled calibration images from the smaller splits share very similar viewpoints. As a result, DUSt3R predictions across view pairs are of similar quality, and global optimization therefore offers limited improvements in point estimation accuracy. Reduced training set diversity and limited scene observation further contribute to the performance decrease.

Therefore, we recommend capturing calibration images from different viewpoints to ensure effective self-calibration.

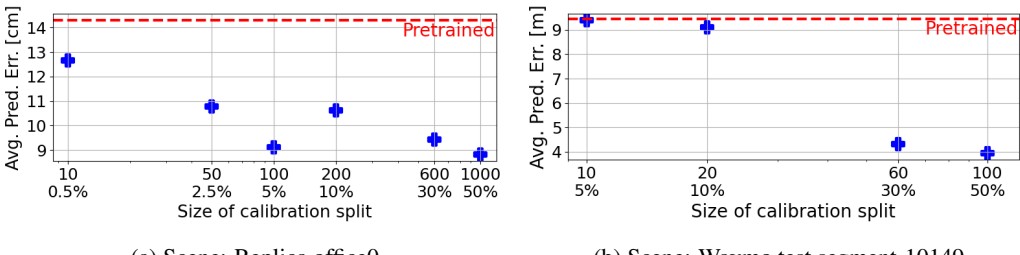

(a) Scene: Replica office0      (b) Scene: Waymo test segment-10149

Figure 10: Ablation Study: Reducing the calibration split size (e.g., using only the first 0.5% of images) may adversely impact self-calibration performance.

## A.7 CONCURRENT MULTI-SCENE SELF-CALIBRATION

Table 7: Quantitative evaluation of multi-scene concurrent self-calibration

| Training Data | office0 (Ours) | All Replica | All except office0 |
|---|---|---|---|
| Avg. Point Pred. Err. [cm] | **8.83** | 9.64 | 11.64 |
| Accuracy [cm] | 4.43 | **4.24** | 6.11 |
| Completeness [cm] | **6.08** | 7.13 | 8.74 |

We tested our pipeline with multi-scene concurrent self-calibration. Specifically, we fine-tuned the pre-trained DUSt3R using pseudo-labeled data from all Replica test scenes and evaluated the self-

calibrated DUSt3R on the office0 test set. We also experimented with excluding scene office0's pseudo-labeled data from the training data during fine-tuning.

The performance of DUSt3R, self-calibrated on different pseudo labeled data, are presented in Tab. 7. We observe comparable performance between concurrent and scene-specific training. However, excluding the target scene data led to a decline in performance. This highlights the importance of including target scene images for model self-calibration.

### A.8   USE DUST3R TO INITIALIZE COLMAP?

As discussed in Sec. 3.3, the vanilla DUSt3R (i.e., the **DUSt3R-Pretrain** baseline) uses a novel 3D-3D-projection-based global optimization to align the DUSt3R-predicted point maps across views. This natually leads to the question: Can we use the commonly adopted 2D-3D-projection-based optimization to align these point maps? Specifically, is it feasible to use DUSt3R predictions to initialize COLMAP's bundle adjustment (Schonberger & Frahm, 2016) to create a stronger baseline?

The short answer is no.

Theoretically, DUSt3R is trained to predict 3D point maps but not explicit 2D-2D or 2D-3D correspondences. While we could attempt to extract reciprocal pixel-level 2D-2D matches by performing nearest-neighbor matching in the 3D point map space (as discussed in Sec. 3.3 of Wang et al. (2024)), these matches are limited because (1) they are only at the pixel level and (2) they can be incomplete due to potential violations of mutual consistency. Directly using these matches for COLMAP bundle adjustment can lead to inferior reconstruction quality.

Experimentally, we attempted to use the 2D-2D correspondences extracted from DUSt3R predictions to initialize COLMAP. We retrieved the matched pixels' globally aligned point map predictions and computed their median 3D positions to initialize the 2D-3D correspondences for COLMAP's bundle adjustment. As reported in Tab. 8 and Fig. 11, this approach, termed **DUSt3R-COLMAP**, significantly underperforms our baselines such as **DUSt3R-Pretrain** and **MASt3R**.

Table 8: Using DUSt3R to initialize COLMAP (i.e. DUSt3R-COLMAP) significantly underperforms other baselines.

| Methods | office0 | | segment-10084 | |
|---|---|---|---|---|
| | Acc.[cm] ↓ | Comp.[cm] ↓ | ATE[m] ↓ | AFE[%] ↓ |
| DUSt3R-Pretrain | 5.22 | 6.78 | **0.79** | **2.79** |
| **DUSt3R-COLMAP** | 44.36 | 56.66 | 5.59 | 89.51 |
| COLMAP (dense) | **2.61** | 89.87 | Fail | Fail |
| MASt3R | 4.69 | **6.05** | 2.85 | 11.87 |

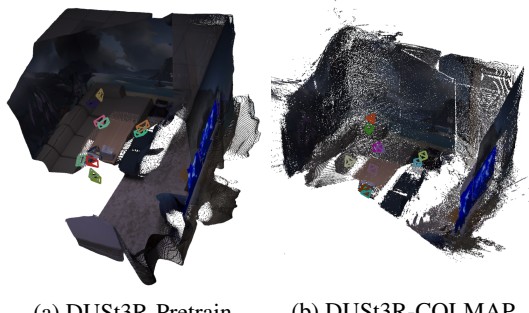

(a) DUSt3R-Pretrain          (b) DUSt3R-COLMAP

Figure 11: Using DUSt3R to initialize COLMAP (i.e. DUSt3R-COLMAP) significantly underperforms the DUSt3R-initialized 3D-projection-based global alignment method (i.e. the DUSt3R-Pretrain baseline).

This observation is consistent with findings in the literature. For example, Leroy et al. (2024) reports (in Tab. 1) that using matches extracted from DUSt3R predictions for map-free localization significantly underperforms MASt3R, which is directly trained for image matching.

A.9 QUALITATIVE COMPARISON AGAINST GROUND TRUTH MESH

We directly compare DUSt3R's multi-view reconstructions with the ground truth mesh to identify and visualize the source of errors.

The qualitative results in Fig. 6 (a, b) demonstrate that our self-calibration method effectively reduces outlier points in DUSt3R reconstructions. However, the outlier points are not the only source of error for the pre-trained DUSt3R model, despite the otherwise visually impressive reconstruction.

In Fig. 12, we directly compare the reconstructions against the ground truth mesh model for the same Replica office0 scene. Beyond the visually obvious improvements highlighted in the green boxes, the self-calibrated model also more accurately captures the geometry of the scene. As a result, after multi-view alignment, the calibrated DUSt3R's reconstruction aligns more closely with the ground truth in various regions, as indicated in the red boxes, while the pre-trained DUSt3R's reconstruction exhibits misalignments in these areas.

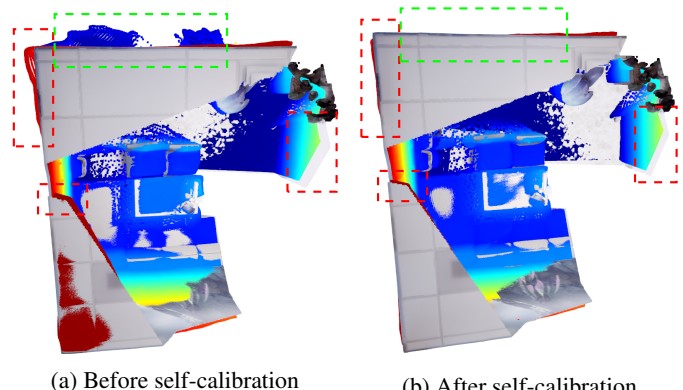

(a) Before self-calibration          (b) After self-calibration

Figure 12: **Qualitative comparison** of DUSt3R multi-view reconstructions before and after self-calibration **against the ground truth mesh model** in the Replica office0 scene (top view for the same reconstructions in Fig. 6 (a,b)). The ground truth mesh, with parts invisible to test camera views culled, is displayed with real vertex colors, while the reconstructions are visualized using heatmap-like colors. Besides the visually obvious improvements in the green boxes, the self-calibrated model also better captures the overall geometry of the scene, demonstrating better alignments with the ground truth at various regions, as indicated in the red boxes.

A.10 REMAINING EXPERIMENTAL RESULTS ON THE REPLICA DATASET

Table 9: Table 2 (continued): Quantitative evaluation of multi-view dense reconstructions on the Replica dataset

| Methods | room0 | | room1 | | room2 | |
|---|---|---|---|---|---|---|
| | Acc. | Comp. | Acc. | Comp. | Acc. | Comp. |
| DUSt3R-Pretrain | 6.97 | 9.97 | 10.54 | 13.13 | 7.79 | 10.92 |
| **DUSt3R-Self-Calib** | 6.83 | 9.86 | **8.88** | 11.54 | 5.80 | 9.10 |
| DUSt3R-GT-FT | 4.59 | 8.15 | 8.64 | 11.35 | 3.35 | 7.34 |
| COLMAP (dense) | 6.30 | 95.02 | 5.54 | 292.35 | 4.88 | 120.52 |
| FlowMap | 66.71 | 226.22 | 96.53 | 140.33 | 37.29 | 274.98 |
| MASt3R | **4.07** | **8.71** | 12.63 | **11.52** | **3.40** | **8.43** |

A.11 FULL EXPERIMENTAL RESULTS ON WAYMO OPEN DATASET

Table 10: **Full quantitative evaluation** results of **multi-view pose estimation** on Waymo Open Dataset (Part 1)

| Methods | segment-10084 | | segment-10149 | | segment-10161 | | segment-10410 | | segment-10488 | |
|---|---|---|---|---|---|---|---|---|---|---|
| | ATE(m) | AFE(%) | ATE(m) | AFE(%) | ATE(m) | AFE(%) | ATE(m) | AFE(%) | ATE(m) | AFE(%) |
| DUSt3R-Pretrain | 0.79 | 2.19 | 0.84 | 3.08 | 0.38 | 0.54 | 0.02 | 0.15 | 0.00 | 0.55 |
| **DUSt3R-Self-Calib** | 0.37 | 0.61 | 0.25 | 2.14 | 0.38 | 0.52 | 0.02 | 0.15 | 0.00 | 0.85 |
| DUSt3R-GT-FT | 0.20 | 0.17 | 0.17 | 1.54 | 0.35 | 0.46 | 0.02 | 0.15 | 0.00 | 0.41 |
| Flowmap | 0.31 | 3.97 | 66.62 | 1.80 | 4.74 | 7.02 | 2.84 | 0.00 | 15.05 | 0.00 |
| PoseDiffusion | 19.43 | 25.07 | 16.76 | 49.18 | 7.65 | 13.85 | 0.03 | 10.84 | 0.00 | 1.78 |
| RayDiffusion | 17.34 | 85.65 | 16.91 | 80.69 | 6.40 | 83.73 | 0.09 | 84.44 | 0.00 | 84.49 |
| RelPose++ | 14.80 | - | 16.20 | - | 4.06 | - | 0.08 | - | 0.00 | - |
| MASt3R | 2.85 | 11.87 | 1.35 | 24.92 | 0.36 | 12.96 | 0.00 | 7.65 | 0.00 | 22.87 |

| Methods | segment-10504 | | segment-10534 | | segment-10649 | | segment-10802 | | segment-10940 | |
|---|---|---|---|---|---|---|---|---|---|---|
| | ATE(m) | AFE(%) | ATE(m) | AFE(%) | ATE(m) | AFE(%) | ATE(m) | AFE(%) | ATE(m) | AFE(%) |
| DUSt3R-Pretrain | 0.29 | 2.19 | 0.74 | 0.84 | 0.95 | 2.84 | 0.35 | 1.60 | 0.01 | 0.74 |
| **DUSt3R-Self-Calib** | 0.38 | 1.79 | 0.67 | 0.76 | 0.49 | 2.54 | 0.35 | 1.08 | 0.02 | 0.94 |
| DUSt3R-GT-FT | 0.40 | 1.04 | 0.67 | 0.75 | 0.29 | 1.73 | 0.39 | 0.55 | 0.02 | 0.64 |
| Flowmap | 66.65 | 1.48 | 14.83 | 6.24 | 36.44 | 13.74 | 21.16 | 0.44 | 1.88 | 0.06 |
| PoseDiffusion | 16.38 | 2.54 | 13.31 | 1.42 | 20.19 | 2.26 | 13.61 | 23.74 | 1.43 | 32.79 |
| RayDiffusion | 17.03 | 85.26 | 10.73 | 84.73 | 18.59 | 85.09 | 12.77 | 81.44 | 2.38 | 84.38 |
| RelPose++ | 16.88 | - | 13.24 | - | 13.69 | - | 12.92 | - | 2.31 | - |
| MASt3R | 1.16 | 25.16 | 9.91 | 19.41 | 0.65 | 20.53 | 1.26 | 24.75 | 0.15 | 9.00 |

| Methods | segment-10980 | | segment-10998 | | segment-11096 | | segment-11436 | | segment-11672 | |
|---|---|---|---|---|---|---|---|---|---|---|
| | ATE(m) | AFE(%) | ATE(m) | AFE(%) | ATE(m) | AFE(%) | ATE(m) | AFE(%) | ATE(m) | AFE(%) |
| DUSt3R-Pretrain | 0.80 | 1.19 | 0.51 | 0.66 | 0.69 | 1.43 | 0.50 | 1.28 | 0.40 | 2.09 |
| **DUSt3R-Self-Calib** | 0.09 | 0.69 | 0.50 | 0.64 | 0.69 | 1.42 | 0.34 | 0.74 | 0.17 | 1.29 |
| DUSt3R-GT-FT | 0.13 | 0.49 | 0.80 | 0.63 | 0.71 | 1.46 | 0.37 | 0.83 | 0.10 | 0.60 |
| Flowmap | 65.17 | 0.66 | 34.91 | 8.07 | 12.88 | 6.72 | 12.43 | 0.59 | 66.10 | 2.23 |
| PoseDiffusion | 18.19 | 31.04 | 11.51 | 5.42 | 12.09 | 8.91 | 13.74 | 15.61 | 11.11 | 8.02 |
| RayDiffusion | 19.12 | 85.00 | 13.80 | 83.88 | 15.08 | 84.71 | 15.23 | 84.22 | 8.92 | 85.25 |
| RelPose++ | 13.55 | - | 14.07 | - | 13.30 | - | 16.90 | - | 11.89 | - |
| MASt3R | 1.61 | 6.59 | 0.57 | 2.48 | 1.11 | 8.72 | 0.51 | 25.26 | 0.46 | 39.65 |

| Methods | segment-11867 | | segment-11933 | | segment-11987 | | segment-12056 | | segment-12153 | |
|---|---|---|---|---|---|---|---|---|---|---|
| | ATE(m) | AFE(%) | ATE(m) | AFE(%) | ATE(m) | AFE(%) | ATE(m) | AFE(%) | ATE(m) | AFE(%) |
| DUSt3R-Pretrain | 0.20 | 1.41 | 0.00 | 0.76 | 0.39 | 1.30 | 0.52 | 1.58 | 0.35 | 0.97 |
| **DUSt3R-Self-Calib** | 0.18 | 1.77 | 0.00 | 0.85 | 0.33 | 1.28 | 0.32 | 2.32 | 0.35 | 1.00 |
| DUSt3R-GT-FT | 4.27 | 2.28 | 0.00 | 0.51 | 0.33 | 1.28 | 0.30 | 0.70 | 0.31 | 0.59 |
| Flowmap | 10.73 | 2.62 | 59.68 | 0.00 | 39.77 | 3.15 | 48.50 | 3.39 | 14.49 | 2.40 |
| PoseDiffusion | 2.44 | 5.20 | 0.00 | 15.37 | 11.22 | 11.04 | 9.41 | 34.37 | 7.24 | 15.39 |
| RayDiffusion | 6.08 | 85.63 | 0.00 | 83.92 | 11.74 | 84.31 | 17.18 | 85.94 | 6.46 | 85.70 |
| RelPose++ | 2.39 | - | 0.00 | - | 11.01 | - | 16.00 | - | 2.92 | - |
| MASt3R | 1.67 | 10.26 | 0.00 | 14.03 | 3.91 | 5.05 | 2.01 | 19.19 | 0.56 | 11.21 |

| Methods | segment-12537 | | segment-12555 | | segment-12892 | | segment-13034 | | segment-13347 | |
|---|---|---|---|---|---|---|---|---|---|---|
| | ATE(m) | AFE(%) | ATE(m) | AFE(%) | ATE(m) | AFE(%) | ATE(m) | AFE(%) | ATE(m) | AFE(%) |
| DUSt3R-Pretrain | 0.63 | 1.38 | 0.02 | 0.45 | 0.20 | 0.68 | 3.84 | 4.12 | 0.01 | 0.79 |
| **DUSt3R-Self-Calib** | 0.54 | 1.42 | 0.02 | 0.75 | 0.08 | 0.56 | 1.63 | 4.19 | 0.01 | 0.43 |
| DUSt3R-GT-FT | 0.10 | 1.17 | 0.02 | 0.04 | 0.05 | 1.14 | 1.97 | 2.28 | 0.01 | 0.25 |
| Flowmap | 32.50 | 0.62 | 46.88 | 0.01 | 17.30 | 0.40 | 43.23 | 12.63 | 16.67 | 0.05 |
| PoseDiffusion | 15.92 | 10.06 | 0.12 | 6.56 | 4.81 | 37.32 | 41.47 | 3.80 | 0.70 | 10.08 |
| RayDiffusion | 13.73 | 86.47 | 0.11 | 83.48 | 6.39 | 83.62 | 33.72 | 83.64 | 1.95 | 84.32 |
| RelPose++ | 6.56 | - | 0.07 | - | 9.31 | - | 36.88 | - | 1.68 | - |
| MASt3R | 0.42 | 2.11 | 0.00 | 4.99 | 0.24 | 15.56 | 1.59 | 24.94 | 0.13 | 7.47 |

| Methods | segment-13585 | | segment-13732 | | segment-13748 | | segment-13763 | | segment-13781 | |
|---|---|---|---|---|---|---|---|---|---|---|
| | ATE(m) | AFE(%) | ATE(m) | AFE(%) | ATE(m) | AFE(%) | ATE(m) | AFE(%) | ATE(m) | AFE(%) |
| DUSt3R-Pretrain | 0.77 | 1.93 | 8.13 | 4.10 | 0.00 | 1.59 | 0.10 | 0.23 | 1.86 | 2.10 |
| **DUSt3R-Self-Calib** | 0.47 | 0.39 | 6.82 | 3.90 | 0.00 | 3.20 | 0.09 | 0.32 | 1.73 | 1.73 |
| DUSt3R-GT-FT | 0.36 | 0.42 | 3.51 | 1.59 | 0.00 | 1.27 | 0.12 | 0.45 | 1.69 | 1.68 |
| Flowmap | 21.94 | 0.91 | 10.93 | 9.08 | 65.35 | 0.00 | 5.87 | 0.70 | 65.69 | 1.20 |
| PoseDiffusion | 11.12 | 12.22 | 41.83 | 19.75 | 0.00 | 4.26 | 7.52 | 11.11 | 13.90 | 21.86 |
| RayDiffusion | 10.40 | 84.53 | 55.53 | 81.32 | 0.00 | 86.28 | 11.27 | 82.18 | 18.50 | 84.38 |
| RelPose++ | 13.53 | - | 46.03 | - | 0.00 | - | 10.67 | - | 16.10 | - |
| MASt3R | 0.37 | 14.36 | 2.02 | 45.88 | 0.00 | 13.12 | 0.56 | 16.25 | 0.58 | 18.27 |

| Methods | segment-13787 | | segment-13790 | | segment-13849 | | segment-13887 | | segment-13944 | |
|---|---|---|---|---|---|---|---|---|---|---|
| | ATE(m) | AFE(%) | ATE(m) | AFE(%) | ATE(m) | AFE(%) | ATE(m) | AFE(%) | ATE(m) | AFE(%) |
| DUSt3R-Pretrain | 1.91 | 1.88 | 4.48 | 4.15 | 0.27 | 0.99 | 1.24 | 2.56 | 0.03 | 0.23 |
| **DUSt3R-Self-Calib** | 0.42 | 0.35 | 3.34 | 5.12 | 0.22 | 0.43 | 0.79 | 2.14 | 0.04 | 0.65 |
| DUSt3R-GT-FT | 0.24 | 0.46 | 1.87 | 1.24 | 0.29 | 0.69 | 0.23 | 1.71 | 0.03 | 0.27 |
| Flowmap | 65.92 | 1.95 | 67.17 | 19.31 | 14.12 | 1.41 | 22.24 | 0.89 | 7.22 | 0.03 |
| PoseDiffusion | 10.28 | 14.95 | 44.05 | 10.49 | 16.78 | 16.87 | 17.17 | 9.37 | 1.40 | 13.21 |
| RayDiffusion | 5.54 | 82.41 | 41.05 | 85.28 | 17.90 | 83.32 | 18.03 | 83.37 | 3.20 | 83.80 |
| RelPose++ | 9.65 | - | 49.12 | - | 11.05 | - | 16.24 | - | 2.51 | - |
| MASt3R | 2.27 | 8.68 | 41.31 | 28.24 | 0.91 | 15.96 | 0.60 | 20.04 | 0.22 | 25.37 |

Table 11: **Full quantitative evaluation** results of **multi-view pose estimation** on Waymo Open Dataset (Part 2)

| Methods | segment-14178 | | segment-14188 | | segment-14386 | | segment-14470 | | segment-14586 | |
|---|---|---|---|---|---|---|---|---|---|---|
| | ATE(m) | AFE(%) | ATE(m) | AFE(%) | ATE(m) | AFE(%) | ATE(m) | AFE(%) | ATE(m) | AFE(%) |
| DUSt3R-Pretrain | 0.22 | 1.71 | 0.82 | 0.96 | 0.03 | 0.52 | 0.35 | 1.04 | 0.32 | 1.14 |
| **DUSt3R-Self-Calib** | 0.09 | 0.26 | 0.83 | 0.99 | 0.03 | 0.55 | 0.20 | 1.13 | 0.26 | 0.70 |
| DUSt3R-GT-FT | 0.05 | 0.48 | 0.88 | 0.65 | 0.02 | 0.37 | 0.11 | 1.02 | 0.12 | 0.38 |
| FlowMap | 10.78 | 0.21 | 34.62 | 8.03 | 67.15 | 0.47 | 67.30 | 0.55 | 20.05 | 1.06 |
| PoseDiffusion | 9.39 | 3.32 | 10.25 | 7.19 | 1.41 | 5.46 | 3.95 | 26.92 | 10.92 | 11.94 |
| RayDiffusion | 11.52 | 86.31 | 9.25 | 84.93 | 1.32 | 86.42 | 5.58 | 85.69 | 10.38 | 87.09 |
| RelPose++ | 8.78 | - | 13.53 | - | 1.43 | - | 4.56 | - | 7.85 | - |
| MASt3R | 0.14 | 33.95 | 0.13 | 8.70 | 0.01 | 6.21 | 0.19 | 0.92 | 0.20 | 17.51 |

| Methods | segment-14631 | | segment-14643 | | segment-14737 | | segment-14918 | | segment-15272 | |
|---|---|---|---|---|---|---|---|---|---|---|
| | ATE(m) | AFE(%) | ATE(m) | AFE(%) | ATE(m) | AFE(%) | ATE(m) | AFE(%) | ATE(m) | AFE(%) |
| DUSt3R-Pretrain | 0.02 | 0.38 | 0.02 | 0.05 | 0.05 | 0.50 | 0.64 | 2.07 | 0.13 | 1.54 |
| **DUSt3R-Self-Calib** | 0.02 | 0.12 | 0.03 | 0.26 | 0.04 | 0.19 | 0.51 | 1.31 | 0.14 | 1.67 |
| DUSt3R-GT-FT | 0.01 | 0.43 | 0.01 | 0.36 | 0.03 | 0.78 | 0.17 | 1.05 | 0.12 | 0.64 |
| FlowMap | 54.43 | 0.02 | 34.54 | 0.01 | 62.51 | 0.02 | 12.51 | 0.13 | 65.77 | 0.67 |
| PoseDiffusion | 0.40 | 10.31 | 0.20 | 4.60 | 0.48 | 58.43 | 8.80 | 14.48 | 6.27 | 1.69 |
| RayDiffusion | 0.51 | 80.16 | 0.70 | 84.18 | 1.01 | 85.80 | 9.30 | 86.42 | 10.48 | 83.64 |
| RelPose++ | 0.37 | - | 0.40 | - | 0.99 | - | 9.97 | - | 7.54 | - |
| MASt3R | 0.01 | 4.37 | 0.02 | 24.20 | 0.01 | 7.18 | 0.15 | 17.16 | 0.35 | 33.00 |

| Methods | segment-15370 | | segment-15410 | | segment-15739 | | segment-15865 | | segment-16050 | |
|---|---|---|---|---|---|---|---|---|---|---|
| | ATE(m) | AFE(%) | ATE(m) | AFE(%) | ATE(m) | AFE(%) | ATE(m) | AFE(%) | ATE(m) | AFE(%) |
| DUSt3R-Pretrain | 0.38 | 1.43 | 0.00 | 0.16 | 0.03 | 0.29 | 0.43 | 1.51 | 0.01 | 0.48 |
| **DUSt3R-Self-Calib** | 0.35 | 1.67 | 0.00 | 0.20 | 0.03 | 0.16 | 0.18 | 0.95 | 0.01 | 0.75 |
| DUSt3R-GT-FT | 0.22 | 1.21 | 0.00 | 0.25 | 0.03 | 0.51 | 0.12 | 0.44 | 0.01 | 0.61 |
| FlowMap | 19.65 | 1.07 | 35.24 | 0.00 | 50.23 | 0.02 | 7.26 | 2.27 | 16.66 | 0.04 |
| PoseDiffusion | 11.95 | 12.85 | 0.00 | 12.67 | 1.65 | 7.88 | 15.60 | 29.29 | 0.22 | 39.76 |
| RayDiffusion | 14.86 | 84.60 | 0.00 | 84.32 | 1.50 | 84.64 | 18.44 | 85.29 | 0.22 | 83.85 |
| RelPose++ | 14.61 | - | 0.00 | - | 1.55 | - | 16.29 | - | 0.29 | - |
| MASt3R | 0.14 | 5.21 | 0.00 | 7.27 | 0.02 | 1.18 | 0.54 | 23.49 | 0.01 | 12.81 |

| Methods | segment-16062 | | segment-16367 | | segment-16418 | | segment-16645 | | segment-16721 | |
|---|---|---|---|---|---|---|---|---|---|---|
| | ATE(m) | AFE(%) | ATE(m) | AFE(%) | ATE(m) | AFE(%) | ATE(m) | AFE(%) | ATE(m) | AFE(%) |
| DUSt3R-Pretrain | 14.93 | 3.35 | 0.52 | 1.55 | 0.12 | 0.41 | 0.89 | 3.36 | 3.91 | 2.13 |
| **DUSt3R-Self-Calib** | 3.23 | 3.09 | 0.31 | 0.99 | 0.05 | 0.36 | 0.47 | 1.71 | 7.73 | 4.40 |
| DUSt3R-GT-FT | 1.09 | 1.26 | 0.05 | 0.54 | 0.05 | 0.57 | 0.50 | 0.91 | 10.14 | 2.56 |
| FlowMap | 64.20 | 5.49 | 3.85 | 1.23 | 6.02 | 0.36 | 63.50 | 4.06 | 64.96 | 11.18 |
| PoseDiffusion | 43.39 | 35.25 | 12.32 | 4.96 | 8.39 | 7.83 | 28.05 | 59.33 | 43.68 | 4.94 |
| RayDiffusion | 46.45 | 85.46 | 13.82 | 84.41 | 7.24 | 84.36 | 23.68 | 84.94 | 37.64 | 83.14 |
| RelPose++ | 23.36 | - | 9.12 | - | 9.41 | - | 29.81 | - | 47.79 | - |
| MASt3R | 4.16 | 36.81 | 0.65 | 10.62 | 0.28 | 21.33 | 2.38 | 29.15 | 12.67 | 46.19 |

| Methods | segment-16743 | | segment-16942 | | segment-16951 | | segment-17030 | | segment-17052 | |
|---|---|---|---|---|---|---|---|---|---|---|
| | ATE(m) | AFE(%) | ATE(m) | AFE(%) | ATE(m) | AFE(%) | ATE(m) | AFE(%) | ATE(m) | AFE(%) |
| DUSt3R-Pretrain | 0.42 | 0.92 | 0.00 | 0.70 | 0.09 | 0.48 | 0.55 | 1.96 | 0.24 | 0.47 |
| **DUSt3R-Self-Calib** | 0.40 | 0.82 | 0.00 | 0.54 | 0.09 | 0.53 | 0.52 | 1.58 | 0.28 | 0.62 |
| DUSt3R-GT-FT | 0.12 | 0.90 | 0.00 | 0.92 | 0.22 | 1.05 | 0.62 | 1.38 | 0.18 | 0.38 |
| FlowMap | 3.49 | 0.39 | 23.53 | 0.00 | 20.71 | 0.41 | 66.69 | 1.29 | 24.01 | 0.79 |
| PoseDiffusion | 8.81 | 29.72 | 0.00 | 37.82 | 1.49 | 3.47 | 8.27 | 18.55 | 6.52 | 2.84 |
| RayDiffusion | 6.38 | 82.81 | 0.00 | 83.65 | 3.96 | 84.95 | 12.34 | 86.11 | 8.17 | 85.18 |
| RelPose++ | 8.58 | - | 0.00 | - | 3.13 | - | 11.00 | - | 5.82 | - |
| MASt3R | 0.35 | 17.33 | 0.00 | 27.89 | 0.14 | 23.55 | 0.47 | 37.95 | 0.24 | 16.97 |

| Methods | segment-17136 | | segment-17174 | | segment-17212 | | segment-17262 | | segment-17351 | |
|---|---|---|---|---|---|---|---|---|---|---|
| | ATE(m) | AFE(%) | ATE(m) | AFE(%) | ATE(m) | AFE(%) | ATE(m) | AFE(%) | ATE(m) | AFE(%) |
| DUSt3R-Pretrain | 0.00 | 0.70 | 0.04 | 1.57 | 1.08 | 2.78 | 0.69 | 2.66 | 3.72 | 4.52 |
| **DUSt3R-Self-Calib** | 0.00 | 0.67 | 0.04 | 0.54 | 0.75 | 1.36 | 0.46 | 0.68 | 1.95 | 3.08 |
| DUSt3R-GT-FT | 0.00 | 0.69 | 0.03 | 0.79 | 0.58 | 1.03 | 0.14 | 0.20 | 0.63 | 1.20 |
| FlowMap | 15.73 | 0.00 | 26.55 | 0.01 | 16.44 | 3.00 | 14.32 | 2.08 | 64.86 | 5.68 |
| PoseDiffusion | 0.00 | 15.44 | 0.77 | 22.26 | 18.75 | 10.02 | 18.89 | 28.26 | 40.69 | 0.61 |
| RayDiffusion | 0.00 | 83.01 | 0.87 | 86.80 | 15.09 | 84.30 | 16.71 | 86.99 | 41.94 | 87.07 |
| RelPose++ | 0.00 | - | 0.73 | - | 15.50 | - | 18.62 | - | 38.25 | - |
| MASt3R | 0.00 | 21.45 | 0.02 | 4.85 | 2.83 | 21.88 | 2.09 | 23.93 | 3.24 | 16.06 |

| Methods | segment-17387 | | segment-17595 | | segment-17652 | | segment-17756 | | segment-17792 | |
|---|---|---|---|---|---|---|---|---|---|---|
| | ATE(m) | AFE(%) | ATE(m) | AFE(%) | ATE(m) | AFE(%) | ATE(m) | AFE(%) | ATE(m) | AFE(%) |
| DUSt3R-Pretrain | 0.08 | 0.31 | 2.03 | 3.07 | 0.00 | 0.39 | 0.17 | 0.43 | 1.18 | 4.16 |
| **DUSt3R-Self-Calib** | 0.08 | 0.40 | 0.74 | 2.26 | 0.00 | 0.39 | 0.16 | 0.41 | 0.63 | 2.18 |
| DUSt3R-GT-FT | 0.08 | 0.25 | 0.27 | 1.37 | 0.00 | 0.86 | 0.69 | 0.78 | 0.41 | 2.76 |
| FlowMap | 67.21 | 0.16 | 64.83 | 5.51 | 18.89 | 0.00 | 65.36 | 5.34 | 65.86 | 0.96 |
| PoseDiffusion | 2.41 | 40.89 | 21.37 | 4.92 | 0.00 | 32.09 | 11.53 | 9.27 | 20.60 | 11.24 |
| RayDiffusion | 1.68 | 84.45 | 17.35 | 86.20 | 0.00 | 84.55 | 10.82 | 83.95 | 19.01 | 84.28 |
| RelPose++ | 1.68 | - | 13.92 | - | 0.00 | - | 12.56 | - | 19.40 | - |
| MASt3R | 0.25 | 2.91 | 1.57 | 5.10 | 0.00 | 28.83 | 1.61 | 13.92 | 1.51 | 32.49 |

Table 12: **Full quantitative evaluation** results of **multi-view pose estimation** on Waymo Open Dataset (Part 3)

| Methods | segment-17835 ATE(m) | AFE(%) | segment-18149 ATE(m) | AFE(%) | segment-19363 ATE(m) | AFE(%) | segment-22189 ATE(m) | AFE(%) | segment-22573 ATE(m) | AFE(%) |
|---|---|---|---|---|---|---|---|---|---|---|
| DUSt3R-Pretrain | 0.44 | 1.10 | 0.02 | 0.72 | 0.61 | 1.07 | 0.94 | 2.63 | 0.22 | 0.61 |
| **DUSt3R-Self-Calib** | 0.40 | 0.86 | 0.02 | 0.86 | 0.48 | 0.80 | 0.52 | 2.91 | 0.14 | 0.50 |
| DUSt3R-GT-FT | 0.15 | 1.75 | 0.02 | 0.51 | 0.47 | 0.69 | 3.42 | 2.06 | 0.18 | 0.61 |
| FlowMap | 11.01 | 0.71 | 11.66 | 0.47 | 15.73 | 1.32 | 31.67 | 26.62 | 6.07 | 0.49 |
| PoseDiffusion | 2.14 | 17.48 | 2.04 | 2.62 | 11.46 | 23.42 | 29.46 | 17.82 | 5.45 | 6.25 |
| RayDiffusion | 10.42 | 83.72 | 1.75 | 86.20 | 12.22 | 85.41 | 22.59 | 85.63 | 12.11 | 84.58 |
| RelPose++ | 6.71 | - | 2.26 | - | 11.66 | - | 18.93 | - | 13.62 | - |
| MASt3R | 0.19 | 29.94 | 0.10 | 4.77 | 0.34 | 31.45 | 1.25 | 32.51 | 0.18 | 13.95 |

| Methods | segment-23632 ATE(m) | AFE(%) | segment-23741 ATE(m) | AFE(%) | segment-23839 ATE(m) | AFE(%) | segment-26012 ATE(m) | AFE(%) | segment-27095 ATE(m) | AFE(%) |
|---|---|---|---|---|---|---|---|---|---|---|
| DUSt3R-Pretrain | 0.00 | 0.10 | 0.25 | 0.42 | 0.01 | 1.42 | 3.91 | 3.57 | 0.18 | 1.06 |
| **DUSt3R-Self-Calib** | 0.00 | 0.06 | 0.21 | 0.35 | 0.01 | 1.36 | 2.11 | 2.97 | 0.08 | 0.79 |
| DUSt3R-GT-FT | 0.00 | 0.59 | 0.11 | 0.49 | 0.01 | 1.67 | 2.24 | 1.65 | 0.08 | 0.37 |
| FlowMap | 18.27 | 0.00 | 66.80 | 0.24 | 64.95 | 0.00 | 32.14 | 1.85 | 24.32 | 7.17 |
| PoseDiffusion | 0.00 | 19.19 | 10.14 | 15.66 | 0.02 | 48.88 | 23.48 | 23.16 | 15.16 | 8.79 |
| RayDiffusion | 0.00 | 84.12 | 9.91 | 84.55 | 0.02 | 88.55 | 24.43 | 82.83 | 17.32 | 86.31 |
| RelPose++ | 0.00 | - | 7.39 | - | 0.02 | - | 22.62 | - | 9.57 | - |
| MASt3R | 0.00 | 10.46 | 0.22 | 23.21 | 0.00 | 3.11 | 1.58 | 24.82 | 0.34 | 13.54 |

| Methods | segment-27143 ATE(m) | AFE(%) | segment-27951 ATE(m) | AFE(%) | segment-28306 ATE(m) | AFE(%) | segment-29065 ATE(m) | AFE(%) | segment-29426 ATE(m) | AFE(%) |
|---|---|---|---|---|---|---|---|---|---|---|
| DUSt3R-Pretrain | 0.71 | 1.25 | 0.47 | 0.86 | 0.55 | 0.86 | 0.89 | 1.04 | 0.97 | 0.93 |
| **DUSt3R-Self-Calib** | 0.37 | 0.44 | 0.38 | 0.98 | 0.56 | 0.81 | 0.90 | 1.04 | 0.90 | 0.83 |
| DUSt3R-GT-FT | 0.28 | 0.46 | 0.89 | 0.88 | 0.57 | 0.82 | 0.85 | 1.00 | 0.12 | 1.16 |
| FlowMap | 22.32 | 1.01 | 0.89 | 1.98 | 28.60 | 1.14 | 1.67 | 0.89 | 28.52 | 2.71 |
| PoseDiffusion | 14.82 | 16.66 | 22.07 | 50.30 | 11.30 | 2.71 | 10.64 | 5.15 | 16.49 | 28.52 |
| RayDiffusion | 16.48 | 86.66 | 28.07 | 81.87 | 10.34 | 83.21 | 13.85 | 84.43 | 15.03 | 83.45 |
| RelPose++ | 13.57 | - | 28.01 | - | 7.06 | - | 10.43 | - | 12.15 | - |
| MASt3R | 0.90 | 25.19 | 0.91 | 21.71 | 0.21 | 10.55 | 0.68 | 16.65 | 0.45 | 20.77 |

| Methods | segment-31225 ATE(m) | AFE(%) | segment-32758 ATE(m) | AFE(%) | segment-33285 ATE(m) | AFE(%) | segment-33418 ATE(m) | AFE(%) | segment-34004 ATE(m) | AFE(%) |
|---|---|---|---|---|---|---|---|---|---|---|
| DUSt3R-Pretrain | 0.35 | 0.63 | 0.27 | 0.49 | 0.02 | 0.60 | 11.16 | 4.69 | 5.13 | 3.97 |
| **DUSt3R-Self-Calib** | 0.34 | 0.75 | 0.27 | 0.50 | 0.02 | 0.42 | 2.24 | 7.65 | 3.92 | 1.94 |
| DUSt3R-GT-FT | 0.11 | 0.33 | 0.23 | 0.25 | 0.02 | 0.75 | 2.44 | 4.25 | 2.16 | 4.62 |
| FlowMap | 21.33 | 1.57 | 35.86 | 0.75 | 1.21 | 0.02 | 3.56 | 4.89 | 34.54 | 7.30 |
| PoseDiffusion | 7.91 | 11.04 | 10.16 | 28.77 | 0.09 | 5.18 | 49.91 | 7.05 | 49.33 | 51.08 |
| RayDiffusion | 9.82 | 81.75 | 10.50 | 85.19 | 0.14 | 86.37 | 48.18 | 85.77 | 45.54 | 84.92 |
| RelPose++ | 6.94 | - | 4.63 | - | 0.13 | - | 31.38 | - | 35.44 | - |
| MASt3R | 1.12 | 11.81 | 0.76 | 5.75 | 0.01 | 17.75 | 1.13 | 26.28 | 4.56 | 23.23 |

| Methods | segment-34590 ATE(m) | AFE(%) | segment-34851 ATE(m) | AFE(%) | segment-35106 ATE(m) | AFE(%) | segment-35228 ATE(m) | AFE(%) | segment-36452 ATE(m) | AFE(%) |
|---|---|---|---|---|---|---|---|---|---|---|
| DUSt3R-Pretrain | 5.48 | 2.32 | 0.16 | 0.80 | 2.38 | 3.22 | 0.06 | 0.35 | 0.13 | 0.69 |
| **DUSt3R-Self-Calib** | 3.54 | 2.04 | 0.10 | 0.77 | 2.18 | 3.30 | 0.05 | 0.36 | 0.31 | 0.29 |
| DUSt3R-GT-FT | 1.42 | 1.51 | 0.12 | 0.63 | 2.63 | 2.77 | 0.04 | 0.27 | 0.24 | 0.56 |
| FlowMap | 16.36 | 2.13 | 18.08 | 0.07 | 67.26 | 10.43 | 17.20 | 0.11 | 38.77 | 5.01 |
| PoseDiffusion | 40.33 | 1.40 | 8.82 | 24.36 | 28.10 | 1.47 | 4.25 | 0.55 | 9.53 | 8.47 |
| RayDiffusion | 35.30 | 85.89 | 6.96 | 83.16 | 26.80 | 85.35 | 3.76 | 85.23 | 9.93 | 85.83 |
| RelPose++ | 24.70 | - | 4.37 | - | 25.92 | - | 2.94 | - | 8.43 | - |
| MASt3R | 1.82 | 29.80 | 0.19 | 11.07 | 4.25 | 14.31 | 0.07 | 8.89 | 0.65 | 11.08 |

| Methods | segment-36541 ATE(m) | AFE(%) | segment-39847 ATE(m) | AFE(%) | segment-40081 ATE(m) | AFE(%) | segment-40379 ATE(m) | AFE(%) | segment-40456 ATE(m) | AFE(%) |
|---|---|---|---|---|---|---|---|---|---|---|
| DUSt3R-Pretrain | 1.25 | 3.06 | 19.23 | 5.13 | 1.03 | 2.54 | 5.89 | 3.82 | 20.43 | 1.63 |
| **DUSt3R-Self-Calib** | 0.35 | 1.52 | 2.24 | 3.72 | 0.98 | 1.82 | 0.94 | 2.58 | 8.01 | 2.78 |
| DUSt3R-GT-FT | 0.40 | 0.80 | 1.78 | 1.75 | 0.84 | 1.79 | 3.55 | 3.67 | 23.53 | 3.77 |
| FlowMap | 26.32 | 1.05 | 34.83 | 4.88 | 67.37 | 2.43 | 17.48 | 4.15 | 2.10 | 2.93 |
| PoseDiffusion | 22.26 | 24.81 | 44.65 | 13.89 | 18.08 | 16.18 | 30.71 | 1.73 | 21.00 | 53.26 |
| RayDiffusion | 24.32 | 84.01 | 41.13 | 85.27 | 14.01 | 86.18 | 33.90 | 86.04 | 25.66 | 85.09 |
| RelPose++ | 22.72 | - | 38.22 | - | 15.42 | - | 40.81 | - | 27.25 | - |
| MASt3R | 0.74 | 16.21 | 23.44 | 23.22 | 0.25 | 36.46 | 36.94 | 33.64 | 0.99 | 25.27 |

| Methods | segment-40540 ATE(m) | AFE(%) | segment-41409 ATE(m) | AFE(%) | segment-45934 ATE(m) | AFE(%) | segment-46325 ATE(m) | AFE(%) | segment-49166 ATE(m) | AFE(%) |
|---|---|---|---|---|---|---|---|---|---|---|
| DUSt3R-Pretrain | 0.48 | 1.78 | 0.20 | 1.99 | 0.80 | 2.09 | 0.02 | 0.76 | 0.01 | 0.73 |
| **DUSt3R-Self-Calib** | 0.31 | 1.74 | 0.09 | 1.07 | 0.49 | 0.83 | 0.02 | 0.61 | 0.01 | 1.38 |
| DUSt3R-GT-FT | 0.12 | 1.00 | 0.05 | 1.21 | 0.33 | 0.69 | 0.01 | 0.55 | 0.01 | 0.73 |
| FlowMap | 11.73 | 0.58 | 66.69 | 3.09 | 34.70 | 0.43 | 2.78 | 0.01 | 17.43 | 0.00 |
| PoseDiffusion | 3.53 | 5.69 | 7.54 | 17.58 | 10.57 | 32.32 | 0.70 | 28.94 | 0.00 | 23.81 |
| RayDiffusion | 8.84 | 85.22 | 10.13 | 84.64 | 12.16 | 84.32 | 1.17 | 85.11 | 0.01 | 87.31 |
| RelPose++ | 9.62 | - | 8.69 | - | 10.43 | - | 1.33 | - | 0.01 | - |
| MASt3R | 0.60 | 22.08 | 0.47 | 31.08 | 0.64 | 13.56 | 0.01 | 10.42 | 0.00 | 3.76 |

Table 13: **Full quantitative evaluation** results of **multi-view pose estimation** on Waymo Open Dataset (Part 4)

| Methods | segment-50269 | | segment-50466 | | segment-51547 | | segment-54445 | | segment-55855 | |
|---|---|---|---|---|---|---|---|---|---|---|
| | ATE(m) | AFE(%) | ATE(m) | AFE(%) | ATE(m) | AFE(%) | ATE(m) | AFE(%) | ATE(m) | AFE(%) |
| DUSt3R-Pretrain | 0.02 | 1.40 | 0.72 | 0.92 | 0.46 | 0.78 | 0.00 | 0.79 | 1.08 | 3.25 |
| **DUSt3R-Self-Calib** | 0.02 | 1.38 | 0.63 | 0.74 | 0.49 | 0.98 | 0.00 | 1.25 | 0.48 | 0.51 |
| DUSt3R-GT-FT | 0.03 | 0.72 | 0.23 | 0.49 | 0.20 | 0.94 | 0.00 | 0.48 | 0.14 | 0.95 |
| FlowMap | 0.11 | 0.06 | 0.43 | 0.36 | 7.09 | 3.33 | 0.75 | 0.00 | 64.80 | 2.95 |
| PoseDiffusion | 0.20 | 36.80 | 12.33 | 9.39 | 15.11 | 5.49 | 0.00 | 21.16 | 18.87 | 36.92 |
| RayDiffusion | 1.07 | 82.00 | 12.55 | 83.03 | 17.47 | 84.13 | 0.00 | 85.85 | 19.73 | 84.64 |
| RelPose++ | 0.91 | - | 4.96 | - | 17.53 | - | 0.00 | - | 17.98 | - |
| MASt3R | 0.06 | 6.43 | 0.16 | 16.06 | 2.23 | 21.26 | 0.00 | 6.52 | 0.14 | 16.11 |

| Methods | segment-56382 | | segment-56480 | | segment-56833 | | segment-57643 | | segment-58104 | |
|---|---|---|---|---|---|---|---|---|---|---|
| | ATE(m) | AFE(%) | ATE(m) | AFE(%) | ATE(m) | AFE(%) | ATE(m) | AFE(%) | ATE(m) | AFE(%) |
| DUSt3R-Pretrain | 0.12 | 0.82 | 30.44 | 4.34 | 1.35 | 1.09 | 0.03 | 0.07 | 0.53 | 1.87 |
| **DUSt3R-Self-Calib** | 0.17 | 1.08 | 12.56 | 8.02 | 1.35 | 0.86 | 0.02 | 0.15 | 0.09 | 0.26 |
| DUSt3R-GT-FT | 0.13 | 0.78 | 2.82 | 1.38 | 1.49 | 0.92 | 0.02 | 0.11 | 0.11 | 0.53 |
| FlowMap | 6.23 | 1.00 | 63.96 | 8.24 | 65.94 | 5.93 | 44.05 | 0.01 | 1.65 | 0.41 |
| PoseDiffusion | 2.67 | 2.83 | 40.82 | 20.78 | 7.03 | 16.44 | 1.90 | 44.77 | 18.16 | 14.25 |
| RayDiffusion | 3.86 | 86.03 | 40.23 | 86.00 | 8.04 | 87.57 | 1.45 | 82.73 | 17.15 | 83.94 |
| RelPose++ | 4.89 | - | 40.78 | - | 7.44 | - | 1.65 | - | 10.05 | - |
| MASt3R | 0.19 | 4.77 | 1.56 | 25.34 | 0.43 | 16.68 | 0.04 | 16.29 | 1.34 | 14.41 |

| Methods | segment-59279 | | segment-59934 | | segment-60792 | | segment-61445 | | segment-61743 | |
|---|---|---|---|---|---|---|---|---|---|---|
| | ATE(m) | AFE(%) | ATE(m) | AFE(%) | ATE(m) | AFE(%) | ATE(m) | AFE(%) | ATE(m) | AFE(%) |
| DUSt3R-Pretrain | 1.31 | 2.01 | 0.37 | 1.20 | 0.12 | 0.96 | 0.28 | 1.35 | 0.59 | 0.79 |
| **DUSt3R-Self-Calib** | 0.61 | 1.62 | 0.27 | 1.11 | 0.18 | 0.51 | 0.27 | 1.29 | 0.48 | 0.59 |
| DUSt3R-GT-FT | 0.45 | 1.23 | 0.31 | 0.64 | 0.17 | 0.55 | 0.27 | 1.24 | 0.32 | 0.97 |
| FlowMap | 63.48 | 2.74 | 26.45 | 1.25 | 22.54 | 0.41 | 0.71 | 2.39 | 16.09 | 0.87 |
| PoseDiffusion | 15.22 | 13.01 | 13.24 | 6.33 | 7.29 | 18.23 | 11.56 | 18.07 | 13.57 | 12.09 |
| RayDiffusion | 13.96 | 85.50 | 13.53 | 84.17 | 9.01 | 86.26 | 12.11 | 85.22 | 12.58 | 84.62 |
| RelPose++ | 11.39 | - | 7.70 | - | 9.16 | - | 13.13 | - | 14.89 | - |
| MASt3R | 0.44 | 12.35 | 0.80 | 15.63 | 0.38 | 12.02 | 0.35 | 15.46 | 0.28 | 20.69 |

| Methods | segment-62287 | | segment-62595 | | segment-62783 | | segment-65030 | | segment-68423 | |
|---|---|---|---|---|---|---|---|---|---|---|
| | ATE(m) | AFE(%) | ATE(m) | AFE(%) | ATE(m) | AFE(%) | ATE(m) | AFE(%) | ATE(m) | AFE(%) |
| DUSt3R-Pretrain | 0.07 | 0.28 | 0.81 | 2.52 | 0.87 | 1.64 | 0.50 | 0.68 | 0.76 | 1.21 |
| **DUSt3R-Self-Calib** | 0.08 | 0.51 | 0.57 | 1.33 | 0.54 | 0.38 | 0.49 | 0.59 | 0.48 | 0.74 |
| DUSt3R-GT-FT | 0.08 | 0.51 | 0.13 | 0.68 | 0.19 | 0.56 | 0.43 | 0.31 | 0.45 | 0.80 |
| FlowMap | 12.08 | 0.03 | 34.90 | 1.51 | 62.22 | 0.70 | 16.36 | 1.63 | 65.16 | 12.63 |
| PoseDiffusion | 1.62 | 2.05 | 14.08 | 11.95 | 15.71 | 15.11 | 12.13 | 18.27 | 14.71 | 0.39 |
| RayDiffusion | 2.58 | 83.36 | 19.96 | 84.87 | 14.12 | 82.92 | 11.29 | 81.05 | 13.52 | 85.09 |
| RelPose++ | 2.40 | - | 13.77 | - | 16.00 | - | 13.27 | - | 13.07 | - |
| MASt3R | 0.03 | 11.91 | 0.28 | 18.00 | 1.21 | 4.88 | 1.59 | 7.40 | 1.55 | 11.60 |

| Methods | segment-68627 | | segment-69228 | | segment-72400 | | segment-72478 | | segment-74355 | |
|---|---|---|---|---|---|---|---|---|---|---|
| | ATE(m) | AFE(%) | ATE(m) | AFE(%) | ATE(m) | AFE(%) | ATE(m) | AFE(%) | ATE(m) | AFE(%) |
| DUSt3R-Pretrain | 0.02 | 0.82 | 0.44 | 0.70 | 1.36 | 1.75 | 0.12 | 0.54 | 0.78 | 1.15 |
| **DUSt3R-Self-Calib** | 0.02 | 0.59 | 0.44 | 0.25 | 0.72 | 0.87 | 0.11 | 0.31 | 0.79 | 1.44 |
| DUSt3R-GT-FT | 0.02 | 0.56 | 0.42 | 0.27 | 0.70 | 0.52 | 0.10 | 0.34 | 0.76 | 1.39 |
| FlowMap | 48.95 | 0.01 | 50.22 | 0.31 | 5.72 | 2.21 | 13.51 | 0.37 | 32.77 | 0.98 |
| PoseDiffusion | 0.46 | 6.46 | 10.73 | 5.88 | 26.81 | 12.04 | 8.41 | 4.67 | 13.54 | 16.78 |
| RayDiffusion | 0.88 | 84.37 | 10.80 | 84.93 | 26.08 | 85.30 | 6.97 | 84.42 | 13.61 | 80.67 |
| RelPose++ | 0.87 | - | 10.72 | - | 25.91 | - | 4.53 | - | 6.81 | - |
| MASt3R | 0.02 | 20.55 | 1.86 | 7.86 | 1.90 | 15.61 | 0.47 | 23.87 | 0.16 | 16.75 |

| Methods | segment-75119 | | segment-78443 | | segment-78551 | | segment-78860 | | segment-79252 | |
|---|---|---|---|---|---|---|---|---|---|---|
| | ATE(m) | AFE(%) | ATE(m) | AFE(%) | ATE(m) | AFE(%) | ATE(m) | AFE(%) | ATE(m) | AFE(%) |
| DUSt3R-Pretrain | 1.09 | 0.96 | 4.83 | 4.48 | 0.79 | 1.24 | 0.24 | 1.32 | 0.42 | 0.59 |
| **DUSt3R-Self-Calib** | 0.61 | 0.75 | 4.58 | 7.77 | 0.46 | 1.01 | 0.24 | 1.85 | 0.38 | 0.47 |
| DUSt3R-GT-FT | 0.44 | 0.29 | 2.89 | 3.13 | 0.25 | 0.43 | 0.18 | 1.11 | 0.24 | 0.50 |
| FlowMap | 65.77 | 5.41 | 11.82 | 27.20 | 13.79 | 6.82 | 1.87 | 1.59 | 38.47 | 4.77 |
| PoseDiffusion | 14.40 | 61.44 | 42.60 | 22.56 | 21.04 | 64.68 | 6.77 | 9.84 | 9.60 | 2.92 |
| RayDiffusion | 21.01 | 82.06 | 43.07 | 84.64 | 22.52 | 83.45 | 6.23 | 84.33 | 10.61 | 84.56 |
| RelPose++ | 20.70 | - | 42.96 | - | 21.81 | - | 3.17 | - | 8.36 | - |
| MASt3R | 1.07 | 28.91 | 25.31 | 21.59 | 2.34 | 26.05 | 2.20 | 13.76 | 0.94 | 9.35 |

| Methods | segment-80858 | | segment-81973 | | segment-82293 | | segment-82491 | | segment-85664 | |
|---|---|---|---|---|---|---|---|---|---|---|
| | ATE(m) | AFE(%) | ATE(m) | AFE(%) | ATE(m) | AFE(%) | ATE(m) | AFE(%) | ATE(m) | AFE(%) |
| DUSt3R-Pretrain | 0.92 | 1.39 | 0.01 | 1.08 | 0.05 | 1.08 | 3.79 | 3.24 | 0.43 | 2.07 |
| **DUSt3R-Self-Calib** | 0.64 | 0.69 | 0.01 | 1.11 | 0.04 | 1.14 | 0.98 | 2.19 | 0.33 | 0.44 |
| DUSt3R-GT-FT | 0.68 | 1.15 | 0.01 | 0.64 | 0.03 | 0.62 | 0.26 | 1.06 | 0.12 | 0.48 |
| FlowMap | 51.33 | 2.02 | 26.73 | 0.01 | 12.76 | 0.64 | 22.68 | 9.36 | 60.06 | 0.56 |
| PoseDiffusion | 21.14 | 57.40 | 0.02 | 51.83 | 1.50 | 28.69 | 27.50 | 12.26 | 9.59 | 13.24 |
| RayDiffusion | 21.98 | 82.51 | 0.02 | 85.90 | 1.37 | 86.79 | 25.74 | 84.38 | 10.92 | 85.28 |
| RelPose++ | 22.75 | - | 0.02 | - | 1.28 | - | 16.99 | - | 13.20 | - |
| MASt3R | 1.49 | 23.72 | 0.01 | 10.38 | 0.10 | 9.91 | 0.69 | 25.66 | 0.46 | 14.95 |

Table 14: **Full quantitative evaluation** results of **multi-view pose estimation** on Waymo Open Dataset (Part 5)

| Methods | segment-86232 | | segment-86840 | | segment-86885 | | segment-89208 | | segment-89936 | |
|---|---|---|---|---|---|---|---|---|---|---|
| | ATE(m) | AFE(%) | ATE(m) | AFE(%) | ATE(m) | AFE(%) | ATE(m) | AFE(%) | ATE(m) | AFE(%) |
| DUSt3R-Pretrain | 0.47 | 0.64 | 0.06 | 0.93 | 0.03 | 0.79 | 0.02 | 0.17 | 2.47 | 3.29 |
| **DUSt3R-Self-Calib** | 0.31 | 0.32 | 0.06 | 0.88 | 0.03 | 0.62 | 0.02 | 0.15 | 2.88 | 3.06 |
| DUSt3R-GT-FT | 0.24 | 0.75 | 0.05 | 0.59 | 0.05 | 0.63 | 0.03 | 0.21 | 2.46 | 3.61 |
| FlowMap | 65.21 | 4.62 | 11.42 | 0.23 | 17.86 | 0.08 | 18.83 | 0.19 | 34.40 | 13.61 |
| PoseDiffusion | 5.34 | 6.57 | 3.50 | 3.54 | 1.34 | 12.19 | 3.08 | 5.69 | 20.01 | 8.13 |
| RayDiffusion | 6.76 | 83.67 | 3.94 | 86.41 | 0.90 | 85.31 | 3.02 | 83.67 | 24.47 | 84.09 |
| RelPose++ | 7.36 | - | 2.54 | - | 1.44 | - | 3.87 | - | 16.97 | - |
| MASt3R | 0.63 | 6.81 | 0.09 | 10.78 | 0.08 | 28.73 | 0.42 | 4.88 | 2.29 | 24.34 |

| Methods | segment-91450 | | segment-93509 | | segment-93554 | | segment-95847 | | segment-98068 | |
|---|---|---|---|---|---|---|---|---|---|---|
| | ATE(m) | AFE(%) | ATE(m) | AFE(%) | ATE(m) | AFE(%) | ATE(m) | AFE(%) | ATE(m) | AFE(%) |
| DUSt3R-Pretrain | 0.07 | 0.94 | 0.61 | 1.90 | 0.08 | 0.83 | 0.08 | 0.43 | 0.07 | 0.47 |
| **DUSt3R-Self-Calib** | 0.07 | 1.26 | 0.30 | 0.41 | 0.05 | 0.90 | 0.06 | 0.32 | 0.06 | 0.53 |
| DUSt3R-GT-FT | 0.14 | 0.62 | 0.18 | 0.76 | 0.12 | 0.80 | 0.06 | 0.94 | 0.05 | 0.30 |
| FlowMap | 61.65 | 0.48 | 0.63 | 0.54 | 34.08 | 1.77 | 24.59 | 0.11 | 19.04 | 0.40 |
| PoseDiffusion | 2.88 | 62.13 | 16.87 | 13.16 | 2.36 | 49.54 | 5.72 | 5.45 | 5.33 | 23.84 |
| RayDiffusion | 5.62 | 84.40 | 16.92 | 85.87 | 2.90 | 87.45 | 6.17 | 84.05 | 6.66 | 84.43 |
| RelPose++ | 4.82 | - | 17.12 | - | 1.99 | - | 5.27 | - | 7.02 | - |
| MASt3R | 0.04 | 11.13 | 0.97 | 23.28 | 0.04 | 0.87 | 0.29 | 30.49 | 0.42 | 35.19 |

