# OpenReview forum: "LoRA3D: Low-Rank Self-Calibration of 3D Geometric Foundation models"
_ICLR.cc/2025/Conference — ICLR 2025 Spotlight_

### Official Review · Reviewer_3dh5 · 2024-10-25

**Soundness:** 3
**Presentation:** 3
**Contribution:** 3
**Rating:** 8
**Confidence:** 4

**Summary:**

This paper presents a method for self-calibrating 3D geometric models. The authors utilize a pre-trained model to generate predicted point maps and their corresponding confidence scores for input images. To automatically identify and mitigate overconfident predictions that may adversely affect fine-tuning performance, they re-parameterize the confidence term as an optimizable weight. This approach effectively reduces the influence of multi-view inconsistent predictions during the optimization process. Ultimately, the refined point maps and calibrated confidence scores serve as pseudo-labels for fine-tuning the pre-trained DUSt3R model using low-rank adaptation (LoRA) techniques. The authors achieve performance enhancements while maintaining an acceptable increase in computational costs.

**Strengths:**

1)	The paper has good writing and is easy to follow.
2)	The authors present a simple but effective pipeline to take advantage of the LoRA technique to improve performance of the 3D foundation model in a self-calibration manner.

**Weaknesses:**

The paper does not present major weaknesses.

**Questions:**

1)	Does Figure 3 solely visualize the prediction confidence and error map of the original DUSt3R method? The caption for Figure 3 should be clarified to reduce ambiguity. Figures 3 and 4 are complementary but they are on separate pages. Readers cannot quickly grasp the points of Figure 3 (a) (b) (c).
2)	Minor issues: missing references in Line 817.

---

> ### Author Response · Authors · 2024-11-23
>
> Thank you for the suggestions!
>
> ### 1. Fig. 3 and Fig. 4 formatting and captioning problem
>
> We have improved the caption of Fig. 3 and placed Fig. 3 and 4 on the same page for easier cross referencing.
>
> ### 2. Missing reference
>
> Fixed!

---

### Official Review · Reviewer_wgpq · 2024-11-02

**Soundness:** 2
**Presentation:** 2
**Contribution:** 2
**Rating:** 6
**Confidence:** 4

**Summary:**

This submission proposes a method to finetune large 3D models such as Duster on individual scenes. The finetuned parameters include camera intrinsics, extrinsics, point maps, and scales. To avoid finetuning a large 3D model directly, the method proposes to finetune them with LoRA. As a result, the proposed method improves per-scene reconstruction quality and camera estimation accuracy in an efficient way.

In terms of weaknesses, my primary concerns are: 1) missing a straightforward global optimisation baseline that is also initialised from Duster predictions, and 2) unconvincing comparison with COLMAP and similar works.

Overall, the paper offers an efficient approach to fine-tuning Duster for scene-specific applications, demonstrating notable gains in reconstruction and camera estimation accuracy. However, the unique advantages of this pipeline are not fully clear, especially given the missing global optimization baseline and the somewhat limited comparisons with similar works like COLMAP.

My initial recommendation leans toward a borderline reject, but I am open to adjusting my rating based on further discussion.

**Strengths:**

The paper presents an efficient method to finetune Duster for individual scenes, showing considerable improvements in terms of reconstruction and camera estimation accuracy.

**Weaknesses:**

***Missing baseline.***
A major weakness to me is missing a straightforward baseline, which is running a global optimisation that is similar to classical global bundle adjustment for all camera intrinsics, extrinsics, point maps, and scales, given initialisations from Dusters predictions. This baseline would also avoid finetuning Duster weights. I am curious about a fair comparison with this baseline, in terms of optimisation time, and per-scene optimisation accuracy.

***The comparison with COLMAP and similar works is not convincing.***
The proposed DUSt3R-Self-Calib always has an initial solution from Duster. It would be more fair to other works if they started from Duster results too.

***Writing (not affecting rating).***
* The text between Sec 5 and Sec 5.1 should be better sectioned and formatted. Currently, tasks and baselines are mixed together.
* In Eq. 7, the notation "W" is reused. It's already used for image size HW.

**Questions:**

1. In Fig 3 and Fig 4, how do Duster predicted confidence maps look like?
2. Fig 3d: this fig shows calibrated confidences are smaller than predictions. Two questions here:
    1) Is it always desirable (better) to have a smaller calibrated confidence compared to Duster predicted confidence?
    2) is there any example where the calibrated confidence is larger than Duster predicted confidence, but still leads to good finetuning results in terms of self-calibration and 3d reconstruction accuracy?
3. Writing (not affecting rating):
    1) L181, "... the global point maps can be further re-parameterized via back-propagation" -- What does “re-parameterized via back-propagation" mean?
    2) L222, "second regularization term" -- it seems like there is only one reg term?

---

> ### Author Response · Authors · 2024-11-23
>
> Thank you very much for the comments!
>
> Before addressing the reviewer’s feedback, we would like to clarify two points about our work to ensure a clear and accurate understanding of our paper's scope.
>
> - Our method is in essence a self-supervised learning pipeline, in which fine-tuning is only one step. The primary difference between our method and an “efficient DUSt3R fine-tuning method” is that our approach does not require any ground truth labels. The only input needed for our pipeline is RGB images for the target scene. Our method does not rely on depth/3D point labels, camera calibration information, camera poses, or external priors.
> Meanwhile, we also found our method to be highly data-efficient, requiring only sparse images to specialize the 3D geometric model to target scenes.
>
> - We are updating DUSt3R model weights (using LoRA) but not “fine-tuning” DUSt3R predictions. In our pipeline, as shown in Fig. 2, we use a robust global optimization method to align multi-view DUSt3R predictions to generate pseudo-labeled data to update the DUSt3R model parameters. During the robust optimization process, we optimize the camera parameters, point maps and scales for global alignment, but we don’t directly “fine-tune” them.
>
> Below are our responses to the reviewer’s comments:
>
> ### 1. Missing baseline for global optimization from DUSt3R initialization
>
> DUSt3R-Pretrain (i.e. the original DUSt3R method) is exactly the baseline the reviewer referred to. Please check out Section 3.3 for details on how the original DUSt3R method uses a 3D-3D-projection-based global optimization to align and refine multi-view DUSt3R predictions. We also explained how the optimization variables are initialized from DUSt3R predictions.
>
> We have further clarified this in the Baselines subsection of Sec. 5.
>
> The optimization time of DUSt3R-Pretrain is comparable to our method (DUSt3R-Self-Calib) since both use the same optimization approach. Their accuracy comparisons are reported in Tables 2, 3, 4 and Figures 6, 7.
>
> We also discussed the feasibility of using DUSt3R to initialize the 2D-3D-projection-based bundle adjustment process below.
>
> ### 2. DUSt3R for COLMAP initialization
>
> Please check out the new Appendix 8 for a detailed discussion of using DUSt3R to initialize COLMAP’s bundle adjustment process. We also summarize the key points below.
>
> Using DUSt3R to initialize COLMAP results in a very weak baseline because:
>
> (1) Theoretically, DUSt3R is trained to predict 3D point maps but not explicit 2D-2D or 2D-3D correspondences. While we could attempt to extract reciprocal pixel-level 2D-2D matches by performing nearest-neighbor matching in the 3D point map space (as discussed in Sec. 3.3 of Wang et al. (2024)), these matches are limited because (a) they are only at the pixel level and (b) they can be incomplete due to potential violations of mutual matching consistency. Directly using these matches for COLMAP bundle adjustment can lead to inferior reconstruction quality.
>
> (2) Experimentally, we attempted to use the 2D-2D correspondences extracted from DUSt3R predictions to initialize COLMAP. We retrieved the matched pixels' globally aligned point map predictions and computed their median 3D positions to initialize the 2D-3D correspondences for COLMAP's bundle adjustment. As reported in the new Tab. 8 (copy-pasted below) and new Fig. 11, this approach, termed \textbf{DUSt3R-COLMAP}, significantly underperforms our baselines such as \textbf{DUSt3R-Pretrain} and \textbf{MASt3R}.
>
> (3) This observation is consistent with findings in the literature. For example, Leroy et al. (2024) reports (in Tab. 1) that using matches extracted from DUSt3R predictions for map-free localization significantly underperforms MASt3R, which is directly trained for image matching.
>
> Table 8: Using DUSt3R to initialize COLMAP (i.e., DUSt3R-COLMAP) significantly underperforms other baselines
>
> |          | **office0**       |       | **segment-10084**     |   |
> |----------|-------------------|-----------------|------------------------|-----------------|
> | **Methods** | **Acc.[cm] ↓**        | **Comp.[cm] ↓**      | **ATE[m] ↓**              | **AFE[%] ↓**        |
> | DUSt3R-Pretrain | 5.22              | 6.78            | **0.79**               | **2.79**         |
> | **DUSt3R-COLMAP** | 44.36             | 56.66           | 5.59                   | 89.51            |
> | COLMAP (dense) | **2.61**          | 89.87           | Fail                   | Fail             |
> | MASt3R            | 4.69              | **6.05**        | 2.85                   | 11.87            |
>
> Therefore, we believe it is not necessary to add this baseline to the Experiments section.
>
> [1] Wang, Shuzhe, et al. "Dust3r: Geometric 3d vision made easy." Proceedings of the IEEE/CVF Conference on Computer Vision and Pattern Recognition. 2024.
>
> [2] Leroy, Vincent, Yohann Cabon, and Jérôme Revaud. "Grounding image matching in 3d with mast3r." European Conference on Computer Vision. Springer, Cham, 2025.

---

> > ### Author Response · Authors · 2024-11-23
> >
> > ### 3. Reformat Sec. 5
> >
> >
> > Done! We have separated the Tasks subsection from the Dataset subsection.
> >
> >
> > ### 4. Notation abuse for W
> >
> > Good catch! We replaced it with \mathcal{W}
> >
> > ### 5. In Fig. 3 and Fig. 4, how do DUSt3R predicted confidence maps look like?
> >
> > The DUSt3R-predicted confidence maps are presented in Fig. 3(b). (Fig. 3 and Fig. 4 use the same example image pair so their predicted confidence maps are the same). We revised the caption for Fig. 3 to clarify that.
> >
> >
> > ### 6. Explain why calibrated confidence is smaller than predicted confidence.
> >
> > First, the absolute value of the calibrated confidence does not matter. We only use the calibrated confidence to threshold point maps for pseudo labeling. As long as they are more correlated with the point estimation errors, which we can clearly observe in Fig. 3(d), we can more easily use this metric to distinguish accurate points from inaccurate points.
> >
> > Second, our confidence update rule, as shown in Eq. 8 (copy-pasted below), determines that the calibrated confidence is ALWAYS smaller than the predicted confidence. So we cannot provide an example of the calibrated confidence being larger than the predicted.
> >
> > \begin{align}
> >     w_p^{v, i} = C_p^{v, i} / (1 + \|e_p^{v, i}\| / \mu)^2
> > \end{align}
> >
> > ### 7. back-propagation (typo)
> >
> > Thanks for catching that. We meant depth back-projection and have corrected it
> >
> > ### 8. 2nd regularization term (typo)
> >
> > Fixed!

---

> ### Comment · Reviewer_wgpq · 2024-11-25
> **Thanks for the detailed response**
>
> Dear Authors,
>
> Thanks for the detailed response and I apologise for misunderstanding some experiment setups. I have changed my rating from 5 to 6.
>
> Best,
> Reviewer wgpq

---

> > ### Author Response · Authors · 2024-11-25
> >
> > Thank you for taking the time to review our updates. We appreciate your efforts! We are glad that we have addressed your concerns. These results aim to provide a more comprehensive insight and strengthen our paper.
> >
> > If you feel that our revisions have sufficiently improved the manuscript, we would be grateful if you could consider further increasing the rating. We are committed to submitting the strongest possible paper and are happy to address any further questions or feedback you may have.

---

> ### Author Response · Authors · 2024-12-02
>
> Dear Reviewer **wgpq**,
>
> Thank you once again for your valuable feedback and your help with our revision! As we approach the end of the rebuttal period, we wanted to double-check if you have any additional questions or concerns regarding our submission. If you feel satisfied with the current state of the paper, we would greatly appreciate your consideration of a stronger endorsement.
>
> Thank you for your time and thoughtful evaluation!
>
> Best regards,
>
> Authors

---

> > ### Comment · Reviewer_3dh5 · 2024-12-02
> >
> > Dear authors,  Thank you for pulling requests and discussions. The paper presents a simple and effective pipeline, and the writing is in good shape. However, considering the technical contribution and level of inspiration, I am not championing it for an award. Best regards

---

> ### Author Response · Authors · 2024-12-02
>
> Dear Reviewer **3dh5**,
>
> Thank you for your response! We fully understand that and sincerely appreciate your feedback on this submission.
>
> We also wanted to note that the original response in this thread was post to reviewer **wgpq**. We already deeply appreciate your strong endorsement of our paper : )
>
> Best,
> Authors

---

### Official Review · Reviewer_XhbG · 2024-11-03

**Soundness:** 3
**Presentation:** 3
**Contribution:** 3
**Rating:** 8
**Confidence:** 2

**Summary:**

This paper presents an effective and straightforward calibration method based on LoRA to fine-tune a 3D foundation model for specialization in a specific scene, achieving a notable performance increase of over 80%. Extensive experiments conducted across multiple datasets and downstream tasks, along with comparisons to various baselines, further validate the model's performance.

**Strengths:**

This task is quite intriguing, presenting an efficient and useful technique for calibrating pre-trained 3D models for specific 3D tasks.
The concept of calibration used to be employed in the field of uncertainty estimation to enhance output quality. Bringing calibration into a 3D foundation model presents a novel approach for "making foundation models" perform more effectively or align better with new or specialized tasks.

The techniques sound and are easy to follow. Despite some poor observations (e.g., low light), there are still sufficient quality observations from the same scene that can provide valuable constraints and useful signals to supervise the calibration process.

The focus on confidence is both interesting and beneficial, as it addresses ambiguity and enhances accuracy. Confidence metrics effectively aid in selecting precise components for fine-tuning during experiments, even facilitating dynamic scenarios.

The use of M-estimation for optimization, as opposed to traditional gradient descent, is another noteworthy aspect.

It is encouraging to see further extensions to Mask3R, which demonstrate the generalizability of this method.

The paper provides good details for reproducibility.

The experiments are extensive, covering various 3D downstream tasks and comparing results with state-of-the-art methods in each category. The results indicate that close to “fine-tuning” level performance can be achieved with minimal calibration.

**Weaknesses:**

The system is designed to adapt to a single scene, which appears to be a limitation of the DUSt3R approach.

To evaluate performance, the methodology requires half of the test split data (1,000 images out of 2,000), which may limit practical applications. I am curious about the results if only 5%, 10%, or 30% of the data were available for calibration. Furthermore, it remains unclear what portion of the calibrated images comes from the TUM and Waymo datasets.

Further details are needed regarding the selection of the 161 test scenes. Is this selection consistent with common practices in previously published baselines?

From Figure 6, it is evident that performance before calibration is already visually impressive, with significant improvements primarily noted in the corner regions. While the reported metrics are impressive, it remains unclear how the calibration contributes to more visually sensible performance.

**Questions:**

Please refer to the questions outlined in the weaknesses section, which includes an ablation study on the portion of data used for calibration, details on how the scenes were selected, and clarification on the visually sensible improvements observed.

---

> ### Author Response · Authors · 2024-11-23
>
> Thank you very much for the insightful comments! Your comments have greatly helped us improve the quality and clarity of our paper.
>
> ### 1. Ablation study on the portion of calibration split
>
> Thanks for the great suggestion! This ablation study is important. Through this ablation study, we gain new insights into our method. Details can be found in the new Appendix 6.
>
> We found that the size of the calibration split plays an important role in self-calibration because it affects the diversity of viewpoints among the calibration images. We show in the new Fig. 10 that as we limit the calibration split to only the first 10 to 50 images in a test scene, we observe a decrease in the performance of the self-calibrated model.
>
> This decline occurs primarily because the sampled calibration images from the smaller splits share very similar viewpoints. As a result, DUSt3R predictions across view pairs are of very similar quality, and global optimization therefore offers limited improvements in point estimation accuracy. Reduced training set diversity and limited scene observation further contribute to the performance decrease.
>
> Therefore, we recommend capturing calibration images from different viewpoints to ensure effective self-calibration.
>
> We have also reported the portion of calibration split for the Waymo and TUM datasets in Sec. 5 and Sec. A.4.
>
> ### 2. Details on how the scenes were selected
>
> Thank you very much for pointing this out!
>
> After reviewing our manuscript, we realized that using the word “sampled” in our Abstract and Experiment sections might imply that we manually selected the test scenes (e.g., "The 161 test scenes are sampled from the …")
>
> In fact, **we used ALL available test scenes from the Replica and Waymo datasets for evaluation.**
>
> We also used the same three TUM scenes that are most frequently tested in SfM and SLAM literature such as ORB-SLAM (Mur-Artal et al. 2015), ElasticFusion (Whelan et al. 2015), NICE-SLAM (Zhu et al. 2022) and Gaussian-Splatting SLAM (Matsuki et al 2024). We did not perform any additional selection or sampling beyond following their practice.
>
> To clarify that, we have removed the word “sampled” and revised the Experiment section to reduce ambiguity.
>
> ### 3. Clarification on the visually sensible improvements
>
> Thank you for the question!
>
> The outlier points as highlighted in Fig. 6 (a) is not the only source of errors. Although the rest of the reconstruction looks impressive, it does not accurately capture the overall scene geometry.
>
> We directly compared the same reconstruction against the ground truth mesh model in the new Fig. 12 (a) to visualize that. Beyond the visually obvious outlier points highlighted in the green boxes. The pre-trained DUSt3R’s reconstruction also fails to align with the mesh in various regions, as marked with the red boxes. This indicates that the pre-trained DUSt3R does not accurately capture the scene geometry, which also significantly contributes to the reported metrics.
>
> As shown in Fig.12 (b), after self-calibration, the DUSt3R reconstruction can more accurately predict the scene geometry, achieving better alignments and higher metric scores.
>
> We provide more detailed discussions of the source of errors in the new Appendix 9.
>
> [1] Mur-Artal, Raul, Jose Maria Martinez Montiel, and Juan D. Tardos. "ORB-SLAM: a versatile and accurate monocular SLAM system." IEEE transactions on robotics 31.5 (2015): 1147-1163.
>
> [2] Whelan, Thomas, et al. "ElasticFusion: Dense SLAM without a pose graph." Robotics: science and systems. Vol. 11. 2015.
>
> [3] Zhu, Zihan, et al. "Nice-slam: Neural implicit scalable encoding for slam." Proceedings of the IEEE/CVF conference on computer vision and pattern recognition. 2022.
>
> [4] Matsuki, Hidenobu, et al. "Gaussian splatting slam." Proceedings of the IEEE/CVF Conference on Computer Vision and Pattern Recognition. 2024.

---

> > ### Comment · Reviewer_XhbG · 2024-11-25
> > **Discussion**
> >
> > Thank you to the authors for their detailed and thoughtful responses to the questions raised. The newly added experiments provide valuable insights and further enhance the potential applications for calibration. Additionally, the inclusion of the new error figure underscores the overall geometric improvements. In summary, I believe this is a good paper that merits acceptance. I will increase my ratings to acceptance.

---

### Official Review · Reviewer_YtdK · 2024-11-08

**Soundness:** 4
**Presentation:** 4
**Contribution:** 4
**Rating:** 8
**Confidence:** 4

**Summary:**

The paper proposes an approach for calibration 3D foundation model, particularly DUSt3R and MASt3R. Given a small sample of images, the paper extracts the most multi-view consistent 3D points (through a robust global optimisation and confidence calibration) and then feeds them to a LoRA-based fine-tune process.

**Strengths:**

The paper is very well written and easy to understand.

I very much appreciate the core calibration framework text section and underlying method, in that is it technically principled, produces strong state-of-the-art improvements, and is presented in an intuitive manor. This is a great example of how ML/CV papers should be written.

It is the first time I see LoRA used in the context of a 3D foundation model. I expect this to help greatly increase the potential impact of the paper.

The overall approach produces significantly improved results over the DUST3R baseline, and the small sizes of the LoRA could potentially lead to practical applications.

**Weaknesses:**

I find little weakness with the paper. Mainly, I would have liked to see more experiments on MASt3R in the body of the main work, and perhaps an inclusion of the calibrated-confidence-based pseudo labelling in the MASt3R integration. This is perhaps the only point that stops me from recommending a strong accept.

**Questions:**

-

---

> ### Author Response · Authors · 2024-11-24
>
> Thank you for the feedback and kind words.
> Due to space constraints, it would be difficult for us to include MASt3R results in the main paper.
> But we will try to add more MASt3R results in Appendix 5 before the rebuttal deadline.

---

> > ### Comment · Reviewer_YtdK · 2024-11-25
> >
> > Thank you for considering to add more MASt3R results.
> >
> > I am of course still convinced the paper should be accepted and am maintainting my rating.

---

> ### Author Response · Authors · 2024-11-27
>
> Thank you again for your feedback! We have added more MASt3R self-calibration results to our manuscript (Appendix 5). Please feel free to let us know if you have further questions!

---

### Meta-Review · Area_Chair_sNiG · 2024-12-18

**Metareview:**

This paper proposes an efficient self-calibration pipeline to specialize the pre-trained models to target scenes using their own multi-view predictions. This paper refines multi-view predictions by robust optimization techniques and incorporates prediction confidence into the geometric optimization process.

The authors generate high-quality pseudo labels for the calibrating views and use low-rank adaptation to fine-tune the models.
The strengths are as below:
1. This paper introduces a regularization term to solve the original optimization problem of multi-view point map alignment.
2. This paper introduces low-rank adaptation (LoRA) to fine-tune the models on the pseudo-labeled data, which achieves the trade-off between performance and efficiency and reduces the catastrophic forgetting of the pre-training data.
3. Experimental results show that great performance improvement on 3D reconstruction, pose estimation and novel-view rendering.
4. This paper is well-motivated from many challenging problems of pre-trained models.
5. This paper is well-written and technically accurate.

Needing more experiments and improving the writing were raised by the reviewers. Please revise the paper according to the discussions before submitting the final version.

**Additional Comments On Reviewer Discussion:**

Reviewer YtdK appreciated the core calibration framework and the application of LoRA in the context of a 3D foundation model. Reviewer  YtdK wanted more experiments on MASt3R. The authors added more MASt3R results in Appendix 5.

Reviewer XhbG thought this work was intriguing and easy to follow. The prediction confidences address ambiguity and enhance accuracy. Further extensions to Mask3R demonstrate the generalizability of the proposed method. Reviewer XhbG asked an ablation study on the portion of data used for calibration , further details regarding the selection of the 161 test scenes and clarification on the visually sensible implementations observed. Authors provided details of ablation study on the portion of calibration split in Appendix 6 and reported the portion of calibration split for the Waymo and TUM datasets in Sec. 5 and Sec A.4. Authors explained the details on how the scenes were selected and removed the word “sampled” to reduce ambiguity. Authors clarified the visually sensible improvements and provided more details in Appendix 9.

Reviewer wgpq concerned the lack of a straight baseline and unconvincing comparisons. Reviewer wgpq thought the unique advantages are not clear. Authors clarified their work in detail, responded that DUSt3R method was the baseline, explained DUSt3R for COLMAP initialization and provided Appendix 8. Authors also revised writing mistakes.

Reviewer 3dh5 considered this work writing well and easy to follow. Reviewer 3dh5 thought authors should clarify the caption of Figure 3. Authors improved the caption of Fig 3.

---

### Decision · Program_Chairs · 2025-01-22

Accept (Spotlight)